# Neuropeptide-mediated synaptic plasticity regulates context-dependent mating behaviors in *Drosophila*

Tianmu Zhang[1], Zekun Wu[1], Yutong Song[1], Tae Hoon Ryu[2,3], Xiaoli Zhang[1], Wenjing Li[1], Yanying Sun[1], Kyle Wong[4], Justine Schweizer[4], Khoi-Nguyen Ha Nguyen[4], Alex Kwan[4], Kweon Yu[2,3], Woo Jae Kim [1,4,5]*

1 HIT Center for Life Sciences, School of Life Science and Technology, Harbin Institute of Technology, Harbin, China, 2 Neurophysiology Research Group, Bio-Nano Research Centre, Korea Research Institute of Bioscience and Biotechnology (KRIBB), Daejeon, Korea, 3 Department of Functional Genomics, Korea University of Science and Technology (UST), Daejeon, Korea, 4 Department of Cellular and Molecular Medicine, University of Ottawa, Ottawa, Canada, 5 Medical and Health Research Institute, Zhengzhou Research Institute of HIT, Zhengzhou, Henan, China

* wkim@hit.edu.cn

## Abstract

Neuropeptides play crucial roles in regulating context-dependent behaviors, but the underlying mechanisms remain elusive. We investigate the role of the neuropeptide SIFa and its receptor SIFaR in regulating two distinct mating duration behaviors in male *Drosophila*: Longer-Mating-Duration (LMD) and Shorter-Mating-Duration (SMD). We found that SIFaR expression in specific neurons is required for both LMD and SMD behaviors. Social context and sexual experience lead to synaptic reorganization between SIFa and SIFaR neurons, altering internal states of brain. We revealed that the SIFa-SIFaR/Crz-CrzR neuropeptide relay pathway is essential for generating distinct interval timing behaviors, with Crz neurons being responsive to the activity of SIFa neurons. Additionally, CrzR expression in glial cell population is critical for regulating LMD behavior. Our study provides insights into how neuropeptides and their receptors modulate context-dependent behaviors through synaptic plasticity and calcium signaling, with implications for understanding the neural circuitry underlying interval timing and neuropeptidergic system modulation of behavioral adaptations.

## Introduction

Plasticity in animal behavior is dependent on the capacity to integrate external stimuli and internal states from a fluctuating environment and, as a result, to modify activity in neuronal circuits of the brain [1,2]. In *Drosophila*, many of the neural systems that are known to influence behavioral circuits in a state-dependent manner and create switches between behaviors make use of various neuropeptides or biogenic amines like serotonin, dopamine, or octopamine [3–9]. These neuromodulatory circuits or

**Data availability statement:** All relevant data are within the paper and its Supporting information files.

**Funding:** This research was supported by a University of Ottawa Startup grant 602496 to WJK, Startup funds from HIT Center for Life Science to WJK, a University of Ottawa Interdisciplinary Research Group Funding Opportunity (IRGFO stream 1 and 2) grants 148101 and 148747 to WJK, a Natural Sciences and Engineering Research Council of Canada (NSERC) Discovery grant (reference: 211406) to WJK, a University of Ottawa Brain and Mind Research Institute/Center for Neural Dynamics Open call project grant 150950 to WJK, a Mitacs Globalink Research Internship Program grant 17268 to WJK. This research was also supported by the Brain Pool Program of the National Research Foundation in Korea grant ZYM5041911 to WJK, Burroughs Wellcome Fund Collaborative Research Travel Grants (reference: 1017486) to WJK and a NVIDIA Academic Hardware Grant Program to WJK. The funders had no role in study design, data collection and analysis, decision to publish, or preparation of the manuscript.

**Competing interests:** The authors have declared that no competing interests exist.

**Abbreviations:** AG,abdominal ganglion; AHLS,Adult Hemolymph-Like Saline; ATR,all-trans retinal; Brp,Bruchpilot; CCT,chemoconnectome; CNS,central nervous system; Crz,Corazonin; *CS*,*Canton-S*; DILPs,*Drosophila* insulin-like peptides; FB,fan-shaped body; FCA,Fly Cell Atlas; fru,fruitless; GnIHR,gonadotropin inhibitory hormone receptors; LAL,lateral accessory lobe; LMD,longer mating duration; MB,mushroom body; MD,mating duration; MIP,myoinhibitory peptide; OL,optic lobe; PB,protocerebral bridge; PK,pyrokinin; ROI,region of interest; SIP,Super Intermediate Protocerebrum; SMD,shorter mating duration; SOG,sub-esophageal ganglion; SPR,sex peptide receptor; UMI,Unique Molecular Identifier; VFB,Virtual Fly Brain.

pathways are not always hardwired; they depend on paracrine signaling or volume transmission, which is based on non-synaptic release of an amine or neuropeptide [4,10]. Interorgan communication is particularly governed by neuromodulatory signaling, which operates via non-synaptic release of amines or neuropeptides. This enables both local (paracrine/volume transmission) and systemic (hormonal action via circulation) pathways to coordinate physiological responses across tissues [11–13].

Neuropeptide relays represent a fundamental mechanism for energy homeostasis, particularly in gut-brain communication [13–18]. These relays critically depend on non-synaptic transmission, where neuropeptides diffuse broadly to coordinate physiological states across tissues. While extensively documented in mammalian systems (e.g., hypothalamic integration of hunger/satiety signals via stomach, adipose, and pancreas communication [13,19]) and conserved in *Drosophila* metabolic regulation [1,14,20–24], the precise circuitry of central neuropeptidergic relays remains poorly characterized. This gap significantly limits our understanding of how such non-synaptic signaling modulates complex behaviors like decision-making.

The dimension of time is the fundamental basis for an animal's survival. Being able to estimate and control the time between events is crucial for all everyday activities [25]. The perception of time in the seconds-to-hours range, referred to as 'interval timing', is involved in foraging, decision-making, and learning via activation of cortico-striatal circuits in mammals [26]. Interval timing requires entirely different neural mechanisms from millisecond or circadian timing [27–29]. There is abundant psychological research on time perception because it is a universal cognitive dimension of experience and behavioral plasticity. Despite decades of research, the genetic and neural substrates of temporal information processing have not been well established except for the molecular bases of circadian timing [30,31]. Thus, a simple genetic model system to study interval timing is required. Fruit fly mating lasts about 20 min—a duration within the range of interval timing mechanisms. Thus, mating duration (MD) provides a relevant model for studying the neural circuits that govern time perception in *Drosophila*. Understanding these circuits requires insight into the neural and behavioral plasticity underlying interval timing [32–38].

The neuropeptide SIFa is a neuromodulator that demonstrates plasticity on a molecular and behavioral level. SIFa is expressed by four neurons located in the pars intercerebrails (PI), and they are the most widely arborizing peptidergic neurons in *Drosophila* [39]. SIFa is synthesized in four PI neurons and transported to the broad regions of central nervous system (CNS), including central complex via axonal projections [39–43]. SIFa has significant impact on various behaviors including feeding-related behavior [41,44], courtship [40,42], sleep [43,45,46], memory [47], and interval timing [48]. SIFa neurons integrate information through several inputs and form a hub to orchestrate many behaviors. The SIFa neurons receive inputs from peptidergic neurons expressing hugin-pyrokinin (PK)/myoinhibitory peptide (MIP) that mediate hunger/satiety [41], *Drosophila* insulin-like peptides (DILPs) that mediates feeding/fasting rhythm [44] and sleep [43,45]. Thus, it is anticipated that the SIFa neurons would play a crucial role in the context-dependent orchestration of multiple behaviors [4].

The mechanisms by which SIFa signaling coordinates behavior via SIFaR remain incompletely resolved. This gap persists partly because SIFaR expression is distributed widely across diverse neural populations, complicating circuit-pecific dissection. Furthermore, neuropeptide-receptor circuits are inherently less tractable than those of neuropeptide-expressing neurons due to systemic compensation, volume transmission, and pleiotropic signaling modes [10,49–51]. Conventional molecular genetics and synaptic connectomics face inherent limitations in resolving neuropeptide-receptor interactions due to SIFaR's broad expression and non-synaptic signaling dynamics. To overcome these specific challenges in dissecting SIFa-SIFaR mechanisms, we employed targeted applications of these approaches—combining genetic perturbation with synaptic tracing to map functional connectivity within candidate pathways. Here, we report that SIFa controls two alternate interval timing behaviors through long-range neuropeptide relay signaling by SIFaR and other important neuropeptides and transmits the internal states of the male brain into decision-making.

## Results

### Two distinct interval timing behaviors are governed by the adult-specific expression of SIFaR

We have described two distinct male *Drosophila melanogaster* behaviors as a model for investigating the neural circuit principles that determine interval timing. Male fruit flies exhibit the LMD behavior, which is characterized by a prolonged duration of mating after exposure to competing males. This behavior is believed to be an adaptation to male mating competition (S1A Fig) [34,35,37,52–59]. SMD is a behavior in which sexually experienced males manifest a shorter duration of mating. This behavior is believed to be an adaptation that allows male flies to conserve energy by mating quickly then continuing to other activities (S1B Fig) [36].

The SIFaR, the receptor for SIFa, has been linked to circadian rhythm, feeding, courtship, sleep, and memory extinction; however, its functional properties remain poorly understood [40–44,47,60]. When we used a pan-neuronal *elav^c155* driver to knock down *SIFaR* in a neuronal population, both LMD and SMD were disrupted (Fig 1A and 1B) compared to controls (Fig 1C–1F). However, RNA interference-mediated knockdown of SIFaR using a pan-glial *repo-GAL4* driver did not significantly alter either LMD or SMD behaviors (S1C and S1D Fig), suggesting that glial expression of SIFaR is not essential for the regulation of interval timing behaviors. The inclusion of *elav-GAL80*, which suppresses GAL4 activity in a pan-neuronal context, was found to restore both LMD and SMD behaviors when SIFaR was knocked down by a pan-neuronal *elav^c155* driver (Fig 1G and 1H) compared to controls (Fig 1I and 1J). The observed reduction in SIFaR expression, driven by *elav^c155*, is deemed sufficient to induce significant disruptions in LMD and SMD behaviors.

To ensure that RNAi did not have an off-target effect, we tested three independent RNAi strains and found that all three RNAi successfully disrupted LMD/SMD when expressed in neuronal populations. (S1E–S1J Fig). We chose to use the HMS00299 line as *SIFaR-RNAi* for all our experiments because it efficiently disrupts LMD/SMD without *UAS-dicer* expression. Employment of broad drivers, including the *tub-GAL4* and the strong neuronal driver *nSyb-GAL4*, with HMS00299 line consistently results in 100% embryonic lethality (S1O–S1R Fig). This phenotype mirrors the homozygous lethality observed in the *SIFaR^B322* mutant. The efficiency of HMS00299 *SIFaR-RNAi* lines was also validated through quantitative PCR analysis (Fig 1K). The silencing of SIFaR mRNA was achieved at approximately 50% using the HMS00299 knockdown line in combination with *tub-GAL80^ts*, with RNAi induction lasting for three days (bottom diagram in Fig 1K). Notably, the same *tub-GAL4* driver, when used without the *tub-GAL80^ts* combination, resulted in embryonic lethality while still reducing SIFaR mRNA levels by 50% after three days of RNAi induction. This finding suggests that SIFaR knockdown using the HMS00299 line with *GAL4* drivers is likely sufficient to elicit the observed LMD and SMD behaviors. This rationale underscores the effectiveness of our experimental approach and its potential implications for understanding the role of SIFaR in mating behaviors. When we used *tub-GAL80^ts* to knock down SIFaR in all neuronal populations in an adult-specific manner, both LMD and SMD were also impaired compared to temperature control (Compare S1K and S1L Fig with S1M and S1N Fig).

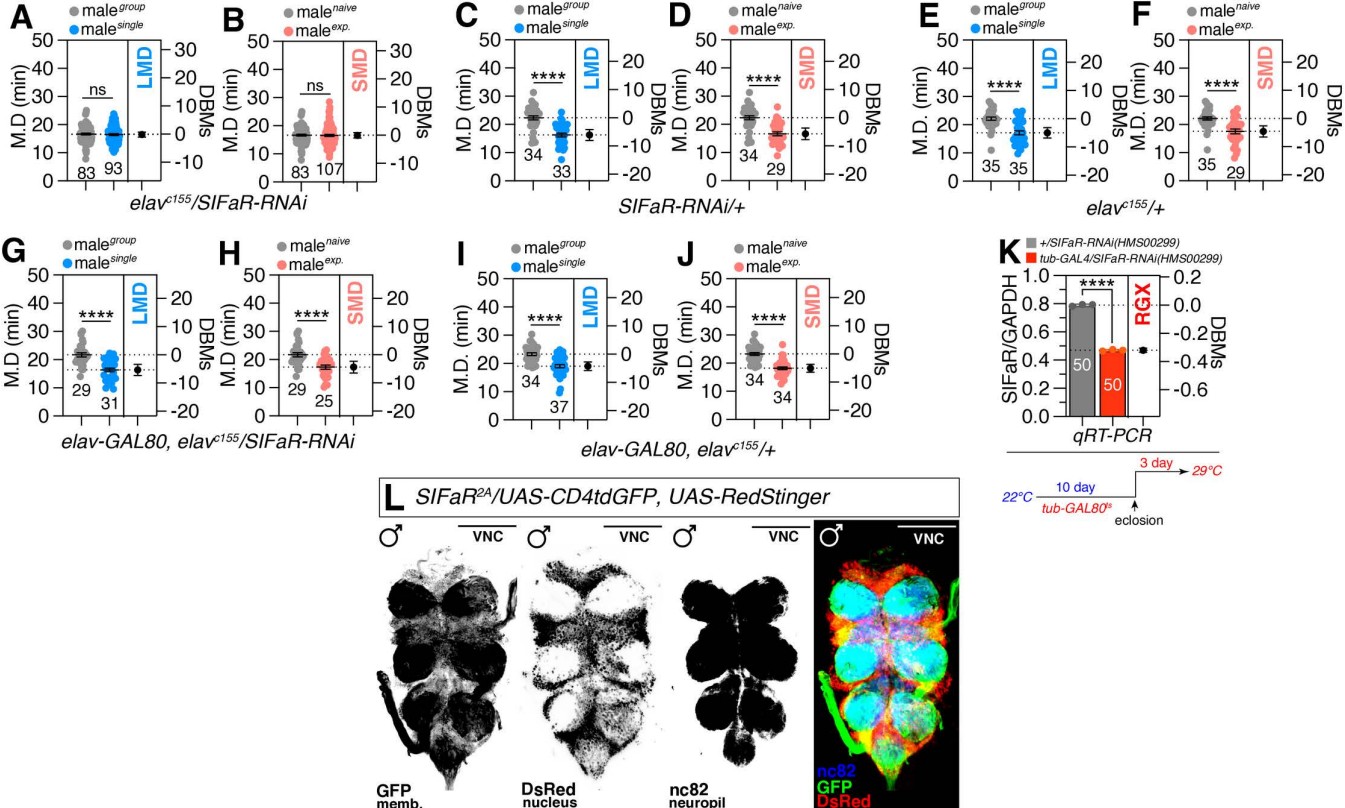

**Fig 1. Interval timing is regulated by adult-specific SIFaR-positive neuronal cells. (A, B)** LMD and SMD assays for *elav^c155^* mediated knockdown of SIFaR *via SIFaR-RNAi.* In the mating duration (MD) assays, light gray data points denote males that were group-reared (or sexually naïve), whereas blue (or pink) data points signify males that were singly reared (or sexually experienced). The dot plots represent the MD of each male fly. The numerical values beneath the dot plots indicate the count of male flies that mated successfully. The mean value and standard error are labeled within the dot plot (black lines). MD represents mating duration. DBMs represent the 'difference between means' for the evaluation of estimation statistics (See Materials and methods). Asterisks represent significant differences, as revealed by the unpaired Student *t* test, and ns represents non-significant differences (*$p < 0.05$, **$p < 0.01$, ***$p < 0.001$, ****$p < 0.0001$). Consequently, data points on graphs marked with asterisks indicate that LMD or SMD behaviors remain unaltered or within normal parameters, whereas those labeled with 'ns' signify that LMD or SMD behaviors have been perturbed due to mutations or genetic alterations in the respective strains. For detailed methods, see the Materials and methods for a detailed description of the MD assays used in this study. In the framework of our investigation, the routine application of internal controls is employed for the vast majority of experimental procedures, as delineated in the "**Mating Duration Assay**" and "**Statistical Tests**" subsections of the Materials and methods section. The identical analytical approach employed for the MD assays is maintained for the subsequent data presented. Sample sizes (*n*) are indicated in the figure panels. **(C–F)** Genetic control assays were performed using heterozygous *SIFaR-RNAi/+* and *elav^c155^/+* males (two-tailed unpaired *t* test). In all plots and statistical tests. Data are presented as mean±s.e.m. ns=not significant ($p > 0.05$), *$p < 0.05$, **$p < 0.01$, ***$p < 0.001$, ****$p < 0.0001$. Sample sizes (*n*) are indicated in the figure panels. **(G, H)** LMD and SMD assays for *elav^c155^*-mediated knockdown of SIFaR *via SIFaR-RNAi* together with *elav-GAL80* (two-tailed unpaired *t* test). Data are presented as mean±s.e.m. ns=not significant ($p > 0.05$), *$p < 0.05$, **$p < 0.01$, ***$p < 0.001$, ****$p < 0.0001$. Sample sizes (*n*) are indicated in the figure panels. **(I, J)** Genetic control assays were performed using heterozygous *elav-GAL80, elav^c155^/+* males (two-tailed unpaired *t* test). In all plots and statistical tests. Data are presented as mean±s.e.m. ns=not significant ($p > 0.05$), *$p < 0.05$, **$p < 0.01$, ***$p < 0.001$, ****$p < 0.0001$. Sample sizes (*n*) are indicated in the figure panels. **(K)** The results of qRT-PCR for SIFaR gene expression. The gray bar represents the control group with the genotype *+/SIFaR-RNAi*. The red bar indicates the experimental group with the genotype *tub-GAL4, tub-GAL80ts/SIFaR-RNAi*. Fifty flies were used in each group. The y-axis depicts the relative expression level of SIFaR, normalized to the CT value of the GAPDH gene. "RGX" denotes relative gene expression. The illustration below depicts the experimental setup where flies were initially reared at 22 °C and then transferred to 29 °C for three days post-eclosion before conducting the qRT-PCR analysis. For detailed methods, see the Materials and methods for a detailed description of the quantitative RT-PCR used in this study. Sample sizes (*n*) are indicated in the figure panels. **(L)** VNC of male flies expressing *SIFaR^2A^* together with *UAS-CD4tdGFP, UAS-Redstinger* were immunostained with anti-GFP (green), anti-DsRed (red), and nc82 (blue) antibodies. The memb. represent cell membrane. The regions within the panel that have undergone immunostaining are clearly delineated. Scale bars represent 100 μm. The panels presented as a gray scale is to clearly show the expression patterns of neurons in brain labeled by *SIFaR^2A^* driver. For detailed methods, see the Materials and methods for a detailed description of the immunostaining procedure used in this study. All images are presented using consistent thresholding across conditions, applied uniformly unless otherwise specified; further details can be found in the Materials and methods section. Underlying data for all graphs can be found in file S1 Data.

We used the newly developed chemoconnectome (CCT) knock-in *SIFaR-GAL4* (*SIFaR^{2A-AC.GAL4}*, *SIFaR^{2A}* hereafter) strain [61] to examine SIFaR expression in the nervous system, while the expression pattern of SIFaR in the brain has been previously characterized [61], our findings reveal abundant SIFaR expression throughout the VNC highly overlapping with neuropil marker nc82 (Fig 1L). Neuropil is a synaptic dense region in the nervous system composed of mostly unmyelinated axons, dendrites, and glial cell processes [62]. The monoclonal antibody nc82, which identifies *Drosophila* Bruchpilot (Brp), a constituent of the CAST1/ERC family and a component of the active zone, has been employed to examine the localization of certain neuronal populations. The significant overlap between the membrane marker of these neurons and the neuropil marker nc82 indicates a possible presence at the active zone [63]; nonetheless, this should not be construed as direct evidence of functionality. The studies demonstrate that SIFaR is extensively expressed in the CNS. Although the expression pattern is indicative, it is crucial to acknowledge that additional tests are necessary to confirm a direct correlation between SIFaR expression in these neurons and the initiation of interval timing behaviors.

**SIFaR expression in specific cell populations is both required and sufficient to maintain interval timing behaviors**

To identify the minimal region labeling GAL4 drivers for testing the functionality of SIFaR-mediated interval timing behaviors, we examined four custom GAL4 drivers [64] targeting the SIFaR regulatory region (Fig 2A). LMD and SMD disappeared when SIFaR-RNAi was co-expressed with *GAL4^{57F10}* (S2E and S2F Fig) and *GAL4^{24F06}* (Fig 2B and 2C) compared to controls (Fig 2D and 2E) but not *GAL4^{23G06}* and *GAL4^{24A12}* (S2A–S2D Fig), indicating that SIFaR expression in these GAL4 driver-expressing cells is required for interval timing behaviors. Targeted reduction of SIFaR expression specifically in the *GAL4^{24F06}* neuronal subset resulted in altered mating behavior. Both singly reared and sexually experienced flies exhibited an extended MD relative to naïve flies, contrary to the expected reduction. This observation indicates a deficit in the neural mechanism responsible for modulating MD, suggesting a disinhibition-like effect within the neural circuitry governing mating behavior. We have also previously observed a similar phenotype when sNPF peptidergic signaling is inhibited in specific neuronal circuits [65]. The prolonged MD observed after *SIFaR* knockdown suggests a disinhibition-like effect within the mating circuitry. This phenotype is similarly evident in *Drosophila* light preference and ethanol sensitization behaviors, indicating a broader regulatory role of *SIFaR* in inhibitory control [66,67].

To determine the cell populations in which SIFaR expression is sufficient to maintain interval timing behaviors, we conducted genetic rescue experiments with the *SIFaR^{B322}* homozygous lethal mutant strain. As shown in Table 1, the expression of *UAS-SIFaR* with *GAL4^{57F10}* and *GAL4^{23G06}* drivers in a *SIFaR^{B322}* mutant background cannot rescue the lethality of the mutant, whereas *GAL4^{24A12}* and *GAL4^{24F06}* can rescue the lethality of the *SIFaR^{B322}* mutant by expressing *UAS-SIFaR*. Among these two GAL4 drivers, only *GAL4^{24F06}* can rescue both lethality and interval timing behaviors in a *SIFaR^{B322}* mutant background by expressing *UAS-SIFaR* (Fig 2F and 2G) compared to controls (Fig 2H–2K), indicating that the cells labeled by *GAL4^{24F06}* driver contain neurons that are critical for both lethality and interval timing behaviors via SIFa-SIFaR neuropeptidergic signaling.

The expression patterns (Fig 2L) and the number of *GAL4^{24F06}* expressing cells (Fig 2M) are similar between male and female brains, these results indicate that *GAL4^{24F06}* labeled cells are highly expressed in the brain and VNC of both sexes. To compare the expression patterns of *GAL4^{24F06}*-positive neurons with those of all other SIFaR-expressing cells, we utilized genetic intersectional methods with a recently developed SIFaR knock-in line, *SIFaR^{2A}* driver (S2G Fig) [61]. The images indicate that the newly identified *GAL4^{24F06}* neurons are pertinent SIFaR-positive cells, with particularly high expression in optic lobe (OL), sub-esophageal ganglion (SOG), and abdominal ganglion (AG) neurons. To better characterize the coexpression of *GAL4^{24F06}* and *SIFaR^{2A}*, we employed an FLP-dependent fluorescence reporter system. These results (S2H Fig) confirm and extend our previous findings, demonstrating consistent overlap between these genetic markers in OL, SOG, and AG.

To further characterize the molecular identity of *GAL4^{24F06}* neuronal population, we first examined potential sexual dimorphism using intersectional approaches with *dsx^{FLP}* and *fru^{FLP}*-based fluorescent reporters. Surprisingly, no *GAL4^{24F06}*

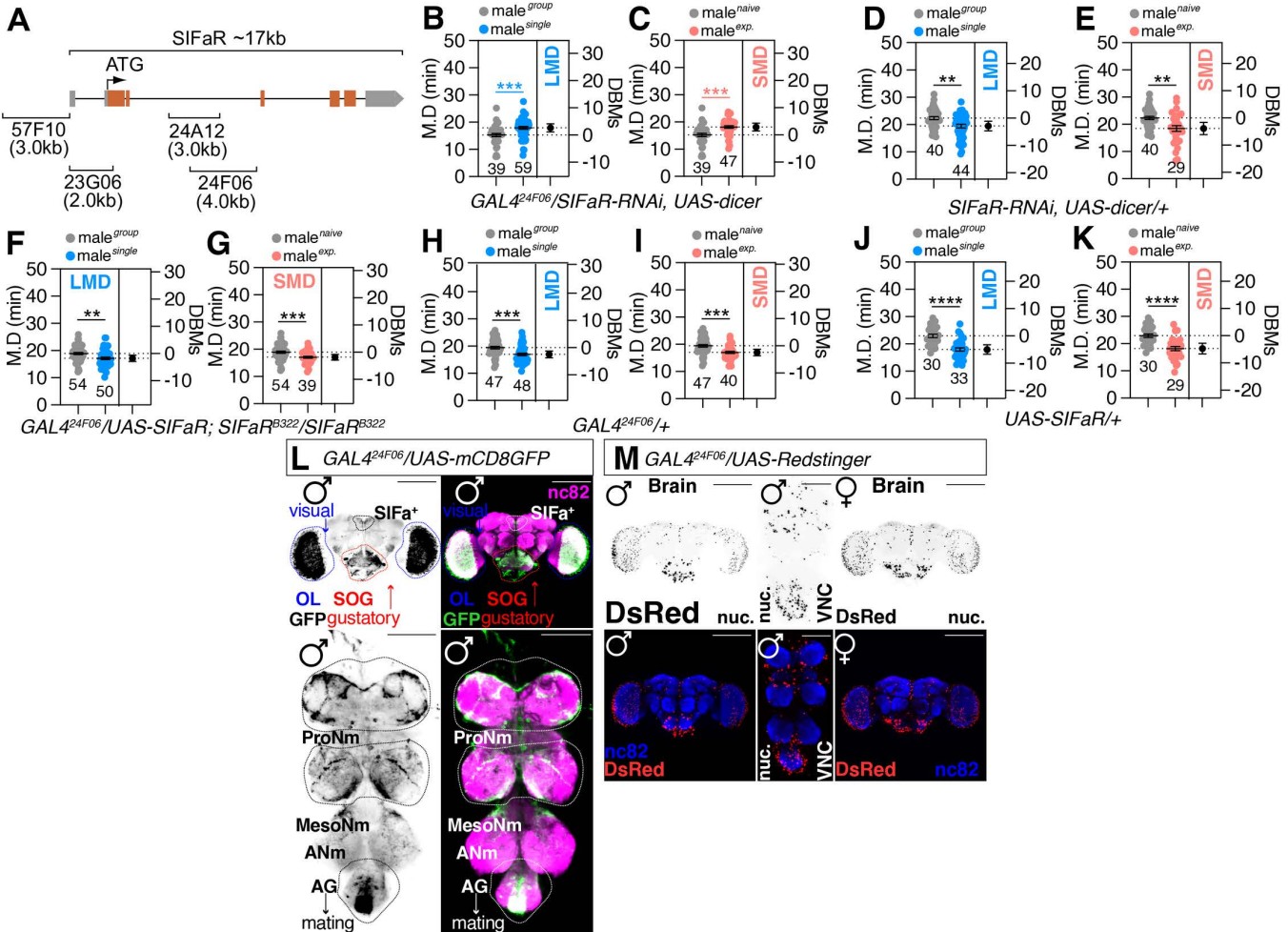

**Fig 2. The expression of *SIFaR24F06* neurons is necessary and sufficient for LMD/SMD. (A)** Diagram of the captured promoter region within each *SIFaR-GAL4* construct. **(B, C)** LMD and SMD assays of *GAL4^24F06* mediated knockdown of SIFaR *via SIFaR-RNAi* together with *UAS-dicer* (two-tailed unpaired *t* test). In all plots and statistical tests. Data are presented as mean±s.e.m. ns=not significant (*p>0.05*), **p<0.05, **p<0.01, ***p<0.001, ****p<0.0001*. Sample sizes (*n*) are indicated in the figure panels. The asterisks (*) indicating statistical significance are marked in blue and pink, representing that the mating duration of this genotype—under both singly reared and sexually experienced conditions—was significantly longer than that of group-reared flies. **(D, E)** Genetic control assays were performed using heterozygous *SIFaR-RNAi, UAS-dicer/+* males (two-tailed unpaired *t* test). In all plots and statistical tests. Data are presented as mean±s.e.m. ns=not significant (*p>0.05*), **p<0.05, **p<0.01, ***p<0.001, ****p<0.0001*. Sample sizes (*n*) are indicated in the figure panels. **(F, G)** Genetic rescue experiments of LMD and SMD assays for *GAL4* mediated overexpression of SIFaR *via GAL4^24F06* in *SIFaR* mutant background flies (two-tailed unpaired *t* test). In all plots and statistical tests. Data are presented as mean±s.e.m. ns=not significant (*p>0.05*), **p<0.05, **p<0.01, ***p<0.001, ****p<0.0001*. Sample sizes (*n*) are indicated in the figure panels. **(H–K)** Genetic control assays were performed using heterozygous *GAL4^24F06/+* and *UAS-SIFaR/+* males (two-tailed unpaired *t* test). In all plots and statistical tests. Data are presented as mean±s.e.m. ns=not significant (*p>0.05*), **p<0.05, **p<0.01, ***p<0.001, ****p<0.0001*. Sample sizes (*n*) are indicated in the figure panels. **(L)** Brain and VNC of male flies expressing *GAL4^24F06* together with *UAS-mCD8GFP*. Scale bars represent 100 μm in brain panels and 50 μm in VNC panels. Dashed circles indicate the region of interest. **(M)** Brain and VNC of male and VNC of female flies expressing *GAL4^24F06* together with *UAS-RedStinger*. Scale bars represent 100 μm in brain panels and 50 μm in VNC panels. Underlying data for all graphs can be found in file S1 Data.

neurons expressed *dsx* or *fru* (S1A and S1B Supplemental information), indicating these neurons lack canonical sexual identity markers. We next mapped neuropeptide coexpression patterns using *lexA^24F06* and *GAL4* lines for peptidergic pathways known to modulate LMD/SMD behaviors (S1 and S2 Tables) [68]. Strikingly, in the SOG and AG, *lexA^24F06* neurons showed extensive colocalization with six neuropeptides: AstA, Capa, FMRFa, Lk, Ms, and Proc (S1C–S1H

**Table 1. Summary of SIFaR^B322 mutant genetic rescue data.**

| GAL4 | Genotype | Lethality (non-TM6B/TM6B) | Courtship | LMD | SMD |
|---|---|---|---|---|---|
| 57F10 | GAL457F10-, SIFaRB322 /UAS-SIFaR; SIFaRB322 | 0/334 | N/A | N/A | N/A |
| 23G06 | GAL423G06, SIFaRB322 /UAS-SIFaR; SIFaRB322 | 0/170 | N/A | N/A | N/A |
| 24A12 | GAL424A12, SIFaRB322 /UAS-SIFaR; SIFaRB322 | 80/90 | No successful mating | N/A | N/A |
| 24F06 | GAL424F06, SIFaRB322 /UAS-SIFaR; SIFaRB322 | 81/90 | Normal | Normal | Normal |

Supplemental information). These findings suggest that: (1) the SOG and AG serve as critical integration hubs where *GAL4^24F06* neurons receive SIFa inputs from the PI region, and (2) these neurons may coordinate interval timing decisions across social-sexual contexts through regulation of these neuropeptidergic pathways.

## Neurons expressing essential SIFaR form functional synapses with SIFa neurons in social context-dependent manner

To determine the neuronal architectures between SIFa neurons and the identified essential SIFaR-expressing neurons, we labeled both SIFa and SIFaR neurons using genetic intersectional methods [69]. As previously reported, SIFa neurons arborize extensively throughout the CNS, but the neuronal processes of *GAL4^24F06*-positive neurons are enriched in the OL, SOG, and AG (GFP signal in Fig 2L). Neuronal processes that are positive for SIFa and SIFaR strongly overlap in the prow (PRW), prothoracic and metathoracic neuromere (ProNm and MesoNm), and AG regions (yellow signals in S3A Fig). We quantified these overlapping neuronal processes between SIFa- and SIFaR-positive neurons and found that approximately 18% of SIFa neurons and 52% of *GAL4^24F06*-positive neurons overlap in brain (S3B and S3C Fig), whereas approximately 48% of SIFa and 54% of *GAL4^24F06*-positive neurons overlap in VNC (S3D and S3E Fig). These findings suggest that SIFa neurons and *GAL4^24F06*-positive neurons form more neuronal processes in the VNC than in the brain. These results imply that *GAL4^24F06*-positive neurons are interconnected in specific regions of the brain and the VNC (S3H Fig). While *GAL4^24F06* labels diverse neurons, the partial overlap with SIFaR highlights specific subregions (e.g., SOG and AG) where SIFa-SIFaR interactions are anatomically plausible, guiding future cell-type-specific investigations.

The observed synaptic alterations in *GAL4^24F06* neurons may arise from the integrated activity of both SIFaR-expressing and non-expressing subpopulations. Nevertheless, given the functional significance of *GAL4^24F06* neurons in rescuing lethality (Table 1) and behavioral deficits (Fig 2F and 2G) in genetic complementation assays, our data support a model wherein SIFa-SIFaR signaling modulates interval timing behaviors through dual mechanisms: direct synaptic modulation and indirect network-level integration.

We utilized synaptobrevin-GRASP (GFP Reconstitution Across Synaptic Partners) [70], a potent activity-dependent marker for synapses in vivo for assessing neuronal connectivity, to examine the synaptic changes between SIFa and SIFaR within brain neurons in various social contexts [71]. Therefore, we employed the SIFa^2A-lexA driver, which specifically labels ventral-posterior SIFa neurons (SIFa^VP) that only project to the brain [48]. Using GRASP experiments, we determined that SIFa^VP-SIFaR neurons form synapses in the gall (GA) of the brain (Fig 3A). GA are made up of a small group of neurons with numerous synapses and relatively few glia around them. They stretch out from the top of the lateral accessory lobe (LAL), adjacent to the ventrolateral protocerebrum and below the spur of the mushroom body (MB) [72]. Since GA are linked to the EB and protocerebral bridge (PB), they may facilitate learning and memory (S4A Fig) [73–75]. We utilized the "Virtual Fly Brain (VFB)" platform, an interactive tool designed for exploring neuronal connectivity, to gain insights into the connectivity of SIFa neurons with four other neurons, specifically GA, FB, and AL (Figs 3B and S4B) [76]. While VFB provides valuable information, it does not offer precise locations of synapses originating from SIFa neurons. To address this limitation, we incorporated data from the FlyEM hemibrain connectome (v1.2.1), which allowed us to confirm

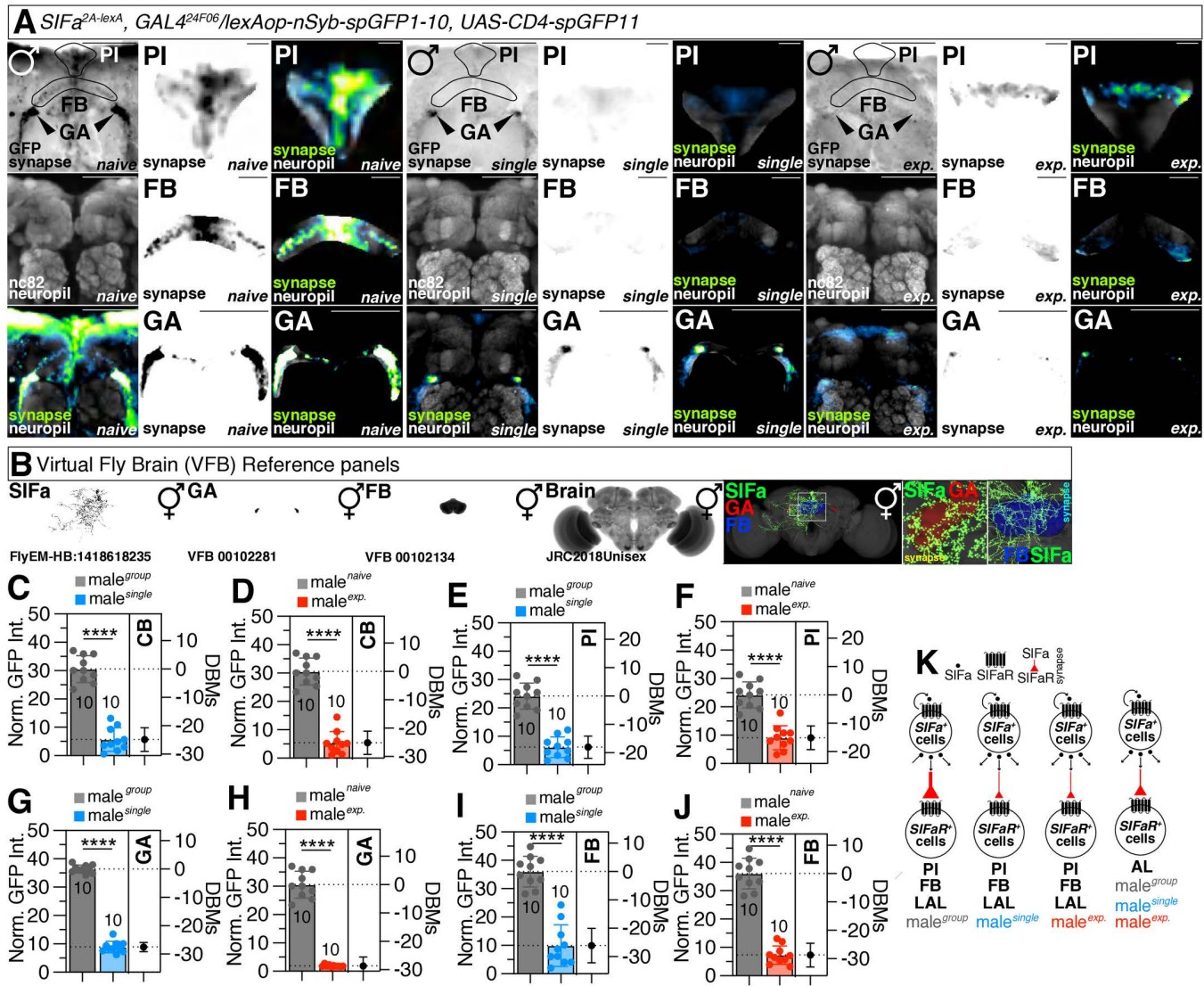

**Fig 3. Social context modulates the formation of synapses between *SIFa2A-lexA* and *GAL424F06* neurons. (A)** GRASP assay for *SIFa²ᴬ⁻ˡᵉˣᴬ* and *GAL4²⁴ᶠ⁰⁶* in PI region (upper panels), FB (middle panels) and GA region (bottom panels) of naïve (left three columns), single (middle three columns), and experienced (right three columns) male flies. Male flies expressing *SIFa²ᴬ⁻ˡᵉˣᴬ, GAL4²⁴ᶠ⁰⁶* and *lexAop-nsyb-spGFP1-10, UAS-CD4-spGFP11* were dissected after 5 days of growth (mated male flies had 1-day of sexual experience with virgin females). Brains of male flies were immunostained with anti-GFP (green) and anti-nc82 (blue) antibodies. The middle panels in each condition are presented as a gray scale to clearly show the synapses connection between *SIFa²ᴬ⁻ˡᵉˣᴬ* and *GAL4²⁴ᶠ⁰⁶*. Circles indicate the regions of interest presented in the panels besides. Synaptic transmission occurs from *SIFa²ᴬ⁻ˡᵉˣᴬ* to *GAL4²⁴ᶠ⁰⁶*. Scale bars represent 50 μm in left columns in each condition, 10 μm PI panels, 25 μm in FB panels, and 50 μm in GA panels. GFP is pseudo-colored as "Green fire blue". **(B)** Expression pattern of *SIFa* in Virtual Fly Brain (VFB) and colocalization between *SIFa* and GA/FB. **(C–J)** Quantification of relative value for synaptic intensity which are formed between *SIFa²ᴬ⁻ˡᵉˣᴬ* and *GAL4²⁴ᶠ⁰⁶* in **(C, D)** CB, **(E, F)** PI, **(G, H)** GA, and **(I, J)** FB between naïve and single male flies. The intensity of GFP fluorescence was normalized to that of the nc82. The same quantification was performed for the relative synaptic intensity in these brain regions between naïve and experienced male flies. The synaptic interactions were visualized utilizing the GRASP system in naïve, single and experienced male flies (two-tailed unpaired $t$ test). In all plots and statistical tests. Data are presented as mean ± s.e.m. ns = not significant ($p > 0.05$), $*p < 0.05$, $**p < 0.01$, $***p < 0.001$, $****p < 0.0001$. Sample sizes ($n$) are indicated in the figure panels. Norm. GFP Int. indicate normalized GFP intensity. See the Materials and methods for a detailed description of the fluorescence intensity analysis used in this study. **(K)** Diagram of various form of connectivity between *SIFa* and *SIFaR* in different sociosexual experience in PI, FB, LAL, and AL. The legend above provides the interpretation for each graphic. Larger graphics indicate a higher number of synapses formed. The subsequent diagrams are identical. Underlying data for all graphs can be found in file S1 Data.

that SIFa projections indeed form actual synapses with GA, AL, FB, and SMP (S3F and S3G Fig) [77]. This multi-faceted approach enhances the robustness of our findings by integrating different data sources to validate neuronal connections.

Males reared in groups (we named naïve condition, hereafter) had stronger connections between SIFa and SIFaR neurons in CB, PI, GA, and fan-shaped body (FB) than males reared in isolation (compare naïve and single in Fig 3C; 3E; 3G; 3I). In addition, the strength of the connections in CB, PI, GA, and FB was reduced in males with sexual experience compared to males without sexual experience (compare naïve and exp. in Fig 3D; 3F; 3H; 3J). Besides, the intensity of synapses decreased when males were socially isolated (S4F and S4G Fig) and sexually experienced (S4I and S4J Fig). However, the size of synapses in different conditions remained constant (S4H and S4K Fig), indicating that the changes of SIFa-SIFaR synapses by sociosexual experiences are authentic. In contrast to synapses between SIFa and SIFaR formed in PI, GA, and FB, synapses between SIFa and SIFaR neurons in AL remained constant or did not vary significantly across sociosexual conditions (Figs 3E–3J; S4C–S4E). In VNC, no synapses were found between *SIFa*[2A-lexA] and *GAL4*[24F06] (S4L Fig). All of these findings indicate that sociosexual experience may have a substantial effect on the connectivity between SIFa and SIFaR neurons (Fig 3K).

## Social context-induced synaptic reorganization precedes calcium-dependent activity modifications in SIFaR neurons and generate feed-forward enhancement

Given the strong expression of SIFaR[2A] cells in the AG of VNC, we employed *GAL4*[SIFa.PT], a GAL4 driver line that project to both AG and the brain area [78], to examine the synaptic connections between SIFa and SIFaR neurons in VNC across various social contexts (Raw data provided in S2 Supplemental information). Social isolation and sexual experiences led to a decrease in the overall synaptic connections between SIFa and SIFaR in the brain and VNC. This reduction was particularly observed in the decline of OL when socially isolated and AG when both socially isolated and sexually experienced (Figs 4A–4G; S5A–S5E). These findings indicate that social isolation leads to a decrease in the SIFa-SIFaR synapses in the visual pathways (OL), as well as a significant decrease in the SIFa-SIFaR synapses in the AG of VNC when males experience social isolation or sexual interactions (Fig 4H). The subtle differences in GRASP signals observed in Fig 3A may stem from the distinct expression patterns of the *SIFa*[2A-lexA] and *GAL4*[SIFa.PT] drivers. We would like to emphasize that the *SIFa*[2A] driver labels only a subset of SIFa neurons in other regions [48].

Next, we investigated the calcium response properties of *GAL4*[24F06] neurons in different social experiences of fly. We analyzed neuronal activity using a CaLexA-based transcriptional reporter system (Raw data provided in S3 Supplemental information) in which sustained neural activity drives GFP expression [79]. We found that the levels of CaLexA signals in both the brain and VNC were significantly elevated in both socially isolated and sexually experienced males when compared to naïve males (S5F–S5J Fig). CaLexA signals originating from *GAL4*[24F06]-positive SOG neurons were found to be at higher levels in males reared singly and with sexual experience than in naïve males (Fig 4I–4K). In comparison to the SOG region, the calcium signals observed in SIFaR[24F06]-positive OL neurons exhibited an increase solely in the singly reared condition, while no significant difference was observed in the sexually experienced condition when compared to the naïve condition (Fig 4L and 4M).

The CaLexA signals originating from the AG region in the VNC showed a decrease when males were subjected to social isolation (Fig 4N). Conversely, these signals exhibited an increase when males had prior sexual experience (Fig 4O). In contrast to the calcium signal changes of AG, the CaLexA signals originating from the VAC region exhibited an increase when males were socially isolated, but no significant differences were observed when sexually experienced (S5K and S5L Fig). These findings suggest a strong link between the synaptic plasticity of SIFa-SIFaR and calcium activity in GAL4[24F06]-positive neurons in males, in response to different social contexts (Fig 4P).

Additionally, SIFa neurons express SIFaR, which enables them to produce SMD behavior [48]. In order to assess the influence of significant alterations in calcium activity that occur precede SIFa-SIFaR synaptic changes, we examined the SIFaR-to-SIFa synaptic changes using tGRASP. No synapses along the SIFaR-to-SIFa direction were observed in

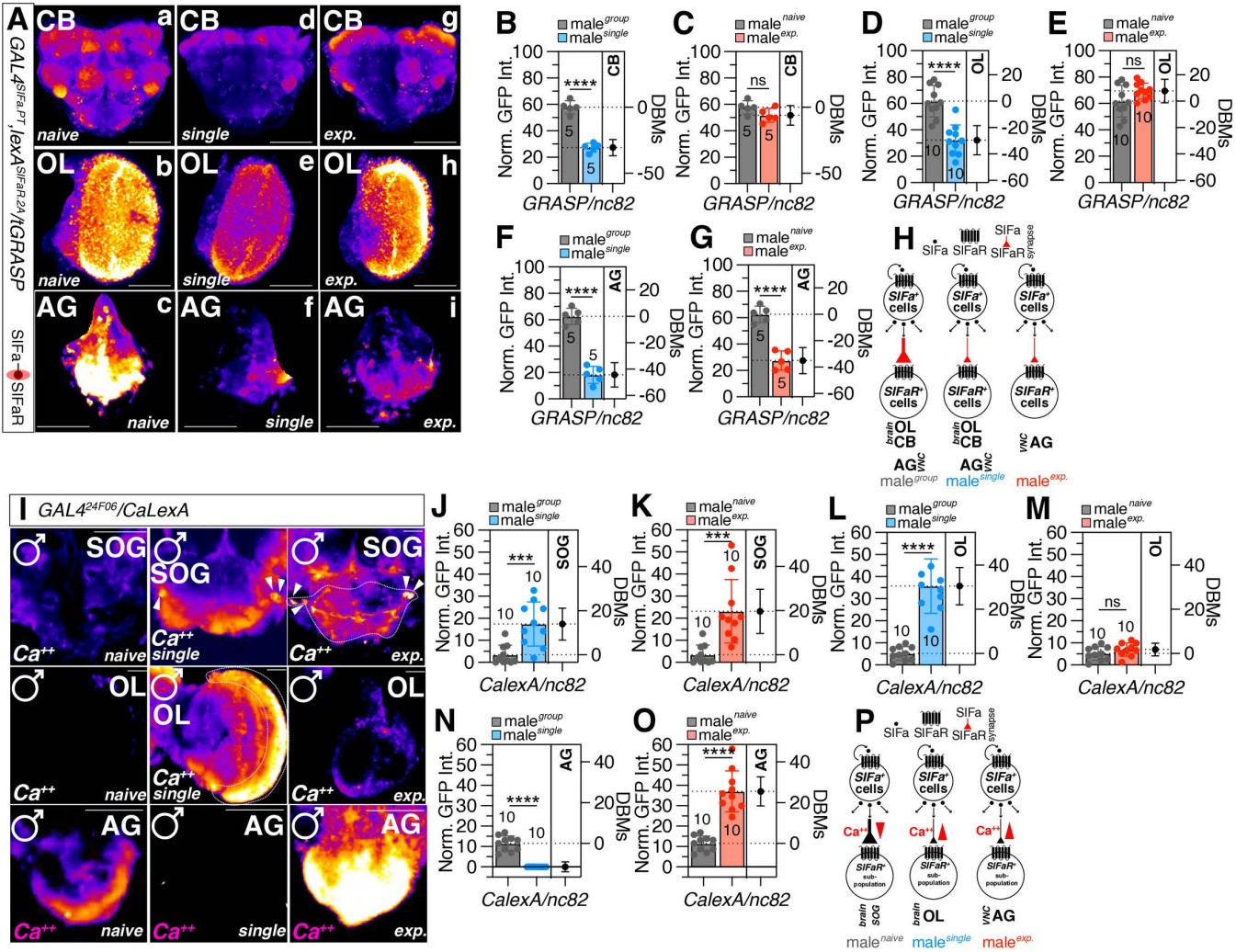

**Fig 4. Social context modulates both the formation of synapses between *GAL4^SIFa.PT* and *lexA^SIFaR-2A* neurons and the calcium-dependent activity of *SIFaR*.** (A) tGRASP assay (resulted in a strong preferential GRASP signal in synaptic regions) for *GAL4^SIFa.PT* and *lexA^SIFaR-2A* in CB, OL, and abdominal ganglion (AG) region of male flies. Male flies expressing *GAL4^SIFa.PT*, *lexA^SIFaR-2A* and *lexAop-2-post-t-GRASP, UAS-pre-t-GRASP* were dissected after 5 days of growth. Synaptic transmission occurs from *GAL4^SIFa.PT* to *lexA^SIFaR-2A*. GFP is pseudo-colored as "red hot". Scale bars represent 100 μm upper panels and 50 μm in AG panels. (B–G) Quantification of relative value for synaptic intensity (two-tailed unpaired *t* test). In all plots and statistical tests. Data are presented as mean ± s.e.m. ns = not significant ($p > 0.05$), *$p < 0.05$, **$p < 0.01$, ***$p < 0.001$, ****$p < 0.0001$. Sample sizes (*n*) are indicated in the figure panels. (H) Schematic representation of the variations in SIFa-SIFaR signaling across distinct regions of the central nervous system (CNS) of male *Drosophila melanogaster* under diverse social contexts. (I) Different levels of neural activity of the brain as revealed by the CaLexA system in naïve, single and experienced flies. Male flies expressing *GAL4^24F06* along with *LexAop-CD2-GFP, UAS-mLexA-VP16-NFAT and LexAop-CD8-GFP-A2-CD8-GFP* were dissected after 5 days of growth (mated male flies had 1-day of sexual experience with virgin females). The dissected brains were then immunostained with anti-GFP (green) and anti-nc82 (blue). GFP is pseudo-colored as "red hot". Boxes indicate the magnified regions of interest presented in the bottom panels. Scale bars represent 50 μm SOG panels, 25 μm in OL panels and AG panels. (J–O) Quantification of relative intensity value for GFP fluorescence (two-tailed unpaired *t* test). In all plots and statistical tests. Data are presented as mean ± s.e.m. ns = not significant ($p > 0.05$), *$p < 0.05$, **$p < 0.01$, ***$p < 0.001$, ****$p < 0.0001$. Sample sizes (*n*) are indicated in the figure panels. (P) Illustration depicting the variations in activity of SIFaR⁺ neurons within various regions of the CNS across different social contexts in male *Drosophila*. The red arrow denotes the calcium ion concentration, with subsequent diagrams following the same representation. Underlying data for all graphs can be found in file S1 Data.

the VNC (S6A–S6C Fig). Significant synaptic plasticity has been observed between SIFaR and SIFa neurons in the PI, PRW, and OL regions of the brain. Additionally, synapses around the PI region also showed an increase in experienced males compared to group-reared males, while a decrease was observed in singly reared males compared to group-reared males (S6D and S6E Fig). The synapses between SIFaR and SIFa in PRW exhibited a decrease both in singly reared males and sexually experienced males (S6F and S6G Fig). The synapses between SIFaR and SIFa in OL exhibited a decrease solely in males who were socially isolated, rather than in those who had sexual experiences (S6H and S6I Fig). Therefore, it can be demonstrated that the synaptic modifications of SIFa-to-SIFaR take place when there are changes in the calcium activity of SIFaR neurons, resulting in changes in SIFaR-to-SIFa synapses (S6J Fig). The evidence indicates that neuronal circuits involving SIFa-to-SIFaR-to-SIFa utilize feed-forward augmentation to modify the internal states of the CNS in response to different social contexts (S6K and S6L Fig).

### The synaptic terminals of *GAL424F06* neurons vary within the CNS of *Drosophila* in different social-sexual contexts to modulate interval timing behavior

To investigate alterations in the synaptic terminals of the $GAL4^{24F06}$-labeled neuron across various social-sexual contexts, we utilized *UAS-sty.eGFP* (Fig 5A). Our findings revealed a significant increase in the synaptic terminals within the OL in the single condition, whereas no differences were observed between the naïve and experienced conditions (Fig 5B and 5C). In the SOG, the synaptic terminal intensity under naïve conditions was lower compared to both the single and experienced conditions (Fig 5D and 5E). In the AG region, the synaptic terminal intensity was higher in the experienced condition than in the naïve condition, with no significant difference observed between the naïve and single conditions (Fig 5F and 5G). Similar modifications in synaptic terminal dynamics have also been noted in the brain and the VNC (S7A–S7D Fig). These results suggest that $GAL4^{24F06}$-labeled neuron in these regions undergoes significant alterations in synaptic terminal dynamics, which are likely involved in modulating interval timing behaviors.

To investigate the response of SIFaR neurons in the AG region following the activation of SIFa neurons in the PI region, we conducted live calcium ($Ca^{2+}$) imaging in the AG region of the VNC, where $SIFaR^{24F06}$ neurons are located (Fig 5H). We tracked $Ca^{2+}$ fluctuations in these neurons using two-photon microscopy in conjunction with a genetically encoded calcium indicator, GCaMP7s, which was expressed under the control of the $GAL4^{24F06}$ driver. To selectively stimulate SIFa neurons, we utilized an optogenetic strategy that involved the red-light-sensitive channelrhodopsin (CsChrimson), expressed using the $lexA^{SIFa.PT}$ driver (Fig 5I). Upon optogenetic activation of SIFa neurons, we noted a significant rise in the activity of $SIFaR^{24F06}$ neurons in the AG region (Fig 5J–5L), as indicated by a prolonged elevation in intracellular $Ca^{2+}$ concentrations that remained at a high level before slowly returning to baseline. This increase in activity was in stark contrast to the control group, which did not receive all-trans retinal (ATR). The results suggest that SIFaR neurons in the AG region respond to the activation of SIFa neurons, which is likely involved in the modulation of interval timing behaviors.

### Corazonin relays the SIFa signaling pathway to SIFaR and CrzR-expressing neurons and glia for the generation of distinct interval timing behavior

The neuropeptide Corazonin (Crz) has been identified as a participant in the regulation of male *Drosophila* MD within the AG of the VNC. The male AG's four neurons that express Crz also express fruitless (fru), and their silencing results in reduced sperm transfer and extended MD [80]. Furthermore, the Crz peptide and Crz-expressing neurons have been characterized as pivotal relay signals in the SIFa-to-SIFaR pathway, which is essential for modulating interval timing behaviors.

Neurons expressing Crz were observed in close proximity to $SIFaR^{24F06}$-expressing neurons within the PRW-SOG of the brain (panels of SOG in Fig 6A). The modulation of interval timing in LMD is primarily facilitated through the visual pathway [34,35]. Given the established absence of Crz peptide in the OL [81], $SIFaR^{24F06}$-expressing neurons within this region may represent a critical neural substrate for LMD behavior that operates independently of Crz signaling. In

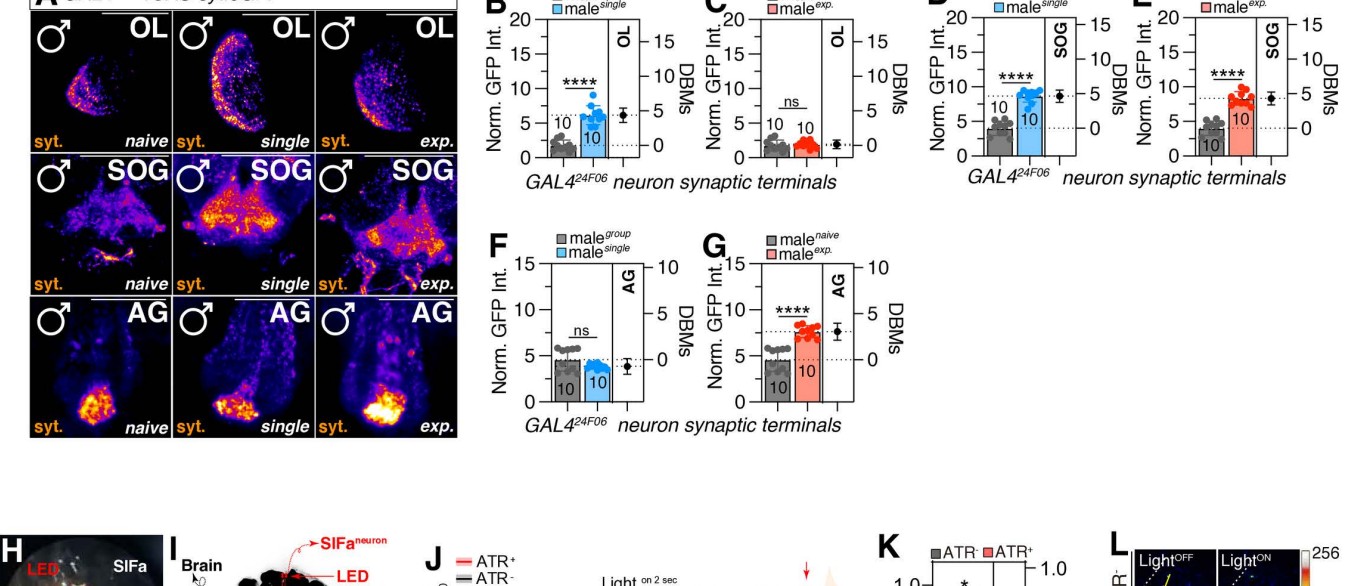

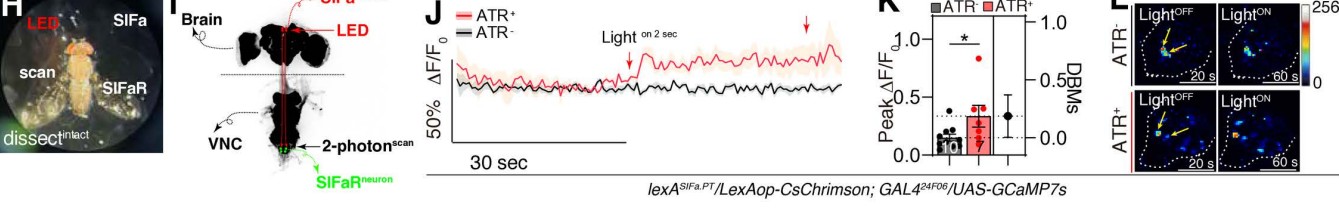

**Fig 5. Social-sexual context altered synaptic terminals of *GAL424F06* to modulate LMD and SMD behaviors. (A)** Male flies brain expressing *GAL4^24F06* together with *UAS-syt.eGFP*. GFP is pseudo-colored as "fire". Scale bars represent 100 μm in OL panels, 50 μm in SOG and AG panels. **(B–G)** Quantification of relative intensity value for GFP fluorescence in OL **(B, C)**, SOG **(D, E)**, and AG **(F, G)** (two-tailed unpaired *t* test). In all plots and statistical tests. Data are presented as mean ± s.e.m. ns = not significant ($p > 0.05$), *$p < 0.05$, **$p < 0.01$, ***$p < 0.001$, ****$p < 0.0001$. Sample sizes (*n*) are indicated in the figure panels. **(H)** An illustrative depiction of a fly immobilized on a plastic plate using UV adhesive on its back. LED light stimulation targeting SIFa neurons in the brain region demarcated by the dashed line. Scanning for neuronal activity was conducted in the area below the dashed line. Original image data by authors. **(I)** Schematic diagram depicting the precise location of the two-photon scanning procedure. Green labels indicate the nucleus of *GAL4^24F06*-labeled neurons located in AG region. Red labels indicate the nucleus of SIFa neurons located in PI region. The fly brain and VNC were immunostained by anti-nc82. **(J–L)** SIFaRs respond to SIFas activation. Fluorescence changes ($\Delta F/F_0$) of GCaMP7s in SIFaRs after optogenetic stimulation of SIFas (two-tailed unpaired *t* test). In all plots and statistical tests. Data are presented as mean ± s.e.m. ns = not significant ($p > 0.05$), *$p < 0.05$, **$p < 0.01$, ***$p < 0.001$, ****$p < 0.0001$. Sample sizes (*n*) are indicated in the figure panels. LED light was fired 2 s after 30 s of dark. Red arrows indicate the time point of light stimulation. *N* = 10 to 9 in each group. Red lines and bars indicate experimental group fed with ATR; black lines and bars indicate control group fed without ATR. See the Materials and methods for a detailed description of the two-photon calcium imaging used in this study. Underlying data for all graphs can be found in file S1 Data.

comparison to the brain, a smaller proportion of SIFaR^24F06 neuron processes exhibited overlap with *Crz-GAL4* in VNC (S7E Fig). However, the overlapping increased in the AG region. The signals of SIFaR^24F06 located outside of the AG region in VNC exhibited minimal overlap with Crz signals (S7E Fig).

By employing the genetic intersectional method, we have determined that approximately 100 neurons in the OL and 2 neurons in Super Intermediate Protocerebrum (SIP) express Crz in conjunction with SIFaR^24F06 neurons (Fig 6B and 6C) [82]. However, the Crz-positive AG neurons in the VNC are not associated with the SIFaR^24F06 driver (S7F Fig), indicating that only Crz neurons in the brain are positive for SIFaR^24F06.

We have previously determined that Crz expression in SIFaR^24F06 cells is required for both LMD and SMD by a systematic screening of *GAL4^24F06*-mediated neuropeptide knockdown (Fig 6D and 6E) compared to controls (Fig 6F and 6G), whereas SIFaR expression in Crz neurons is only required for SMD and not LMD (Fig 6H and 6I) compared to controls

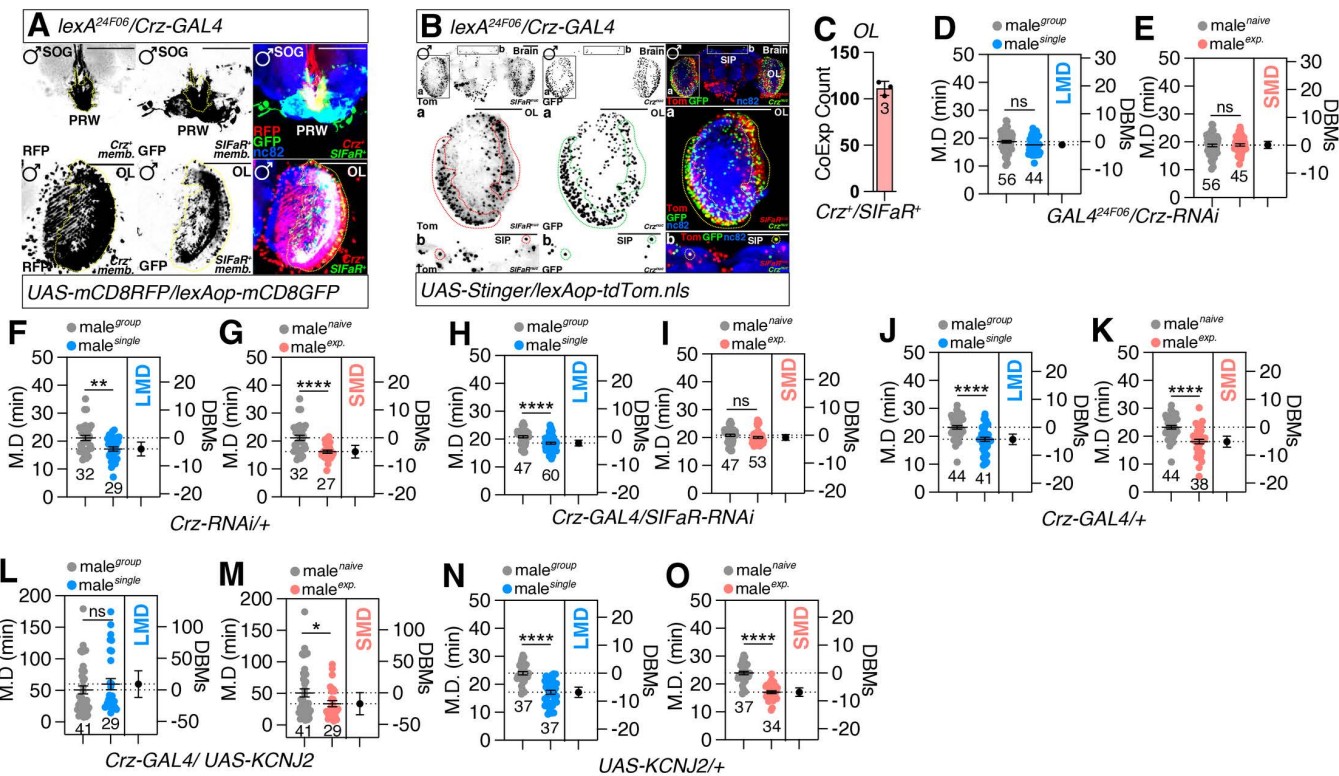

**Fig 6. *Crz* modulate LMD and SMD behaviors through SIFa-SIFaR signaling. (A)** Male flies brain expressing *Crz-GAL4* and *lexA^24F06* drivers together with *UAS-mCD8RFP* and *lexAop-mCD8GFP*. Boxes indicate the magnified regions of interest presented in the bottom panels. Scale bars represent 50 μm. **(B)** Male flies brain and VNC expressing *Crz-GAL4* and *lexA^24F06* drivers together with *UAS-Stinger* and *lexAop-toTomato.nls*. Boxes indicate the magnified regions of interest presented in the bottom panels. Scale bars represent 50 μm. **(C)** Quantification of overlapped cells between *Crz-GAL4* and *lexA^24F06* in OL. Abbreviation: CoExp Count, Coexpression Count. Sample sizes (*n*) are indicated in the figure panel. **(D, E)** LMD and SMD assays for *GAL4^24F06* mediated knock down of Crz *via Crz-RNAi* (two-tailed unpaired *t* test). In all plots and statistical tests. Data are presented as mean ± s.e.m. ns = not significant (*p > 0.05*), **p < 0.05*, ***p < 0.01*, ****p < 0.001*, *****p < 0.0001*. Sample sizes (*n*) are indicated in the figure panels. **(F, G)** Genetic control assays were performed using heterozygous *Crz-RNAi*/+ males (two-tailed unpaired *t* test). In all plots and statistical tests. Data are presented as mean ± s.e.m. ns = not significant (*p > 0.05*), **p < 0.05*, ***p < 0.01*, ***p < 0.001*, ****p < 0.0001*. Sample sizes (*n*) are indicated in the figure panels. **(H, I)** LMD and SMD assays for *Crz-GAL4* mediated knock down of SIFaR *via SIFaR-RNAi* (two-tailed unpaired *t* test). In all plots and statistical tests. Data are presented as mean ± s.e.m. ns = not significant (*p > 0.05*), **p < 0.05*, ***p < 0.01*, ***p < 0.001*, ****p < 0.0001*. Sample sizes (*n*) are indicated in the figure panels. **(J, K)** Genetic control assays were performed using heterozygous *Crz-GAL4*/+ males (two-tailed unpaired *t* test). In all plots and statistical tests. Data are presented as mean ± s.e.m. ns = not significant (*p > 0.05*), **p < 0.05*, ***p < 0.01*, ***p < 0.001*, ****p < 0.0001*. Sample sizes (*n*) are indicated in the figure panels. **(L, M)** LMD and SMD assays for *Crz-GAL4* mediated electrical silencing *via UAS-KCNJ2* (two-tailed unpaired *t* test). In all plots and statistical tests. Data are presented as mean ± s.e.m. ns = not significant (*p > 0.05*), **p < 0.05*, ***p < 0.01*, ***p < 0.001*, ****p < 0.0001*. Sample sizes (*n*) are indicated in the figure panels. **(N, O)** Genetic control assays were performed using heterozygous *UAS-KCNJ2*/+ males (two-tailed unpaired *t* test). In all plots and statistical tests. Data are presented as mean ± s.e.m. ns = not significant (*p > 0.05*), **p < 0.05*, ***p < 0.01*, ***p < 0.001*, ****p < 0.0001*. Sample sizes (*n*) are indicated in the figure panels. Underlying data for all graphs can be found in file S1 Data.

(Fig 6J and 6K) [68]. It has been reported that KCNJ2, which functions by inwardly rectifying the K⁺ channel to impede Crz neuronal activity, can substantially extend the duration of mating [80]. Previous research has documented a significant augmentation in the MD exhibited by these males. However, this genetic manipulation only disrupted LMD and not SMD (Fig 6L and 6M) compared to controls (Fig 6N and 6O).

To elucidate the direct response of Crz neurons to the activity of SIFa neurons, we conducted live calcium (Ca²⁺) imaging in the SIP, OL, and AG region of the VNC, where Crz neurons are situated (Figs 7A, 7B, and 7E; S7G). Upon optogenetic stimulation of SIFa neurons, we observed a tendency to maintain the activity of Crz neurons in OL and AG regions

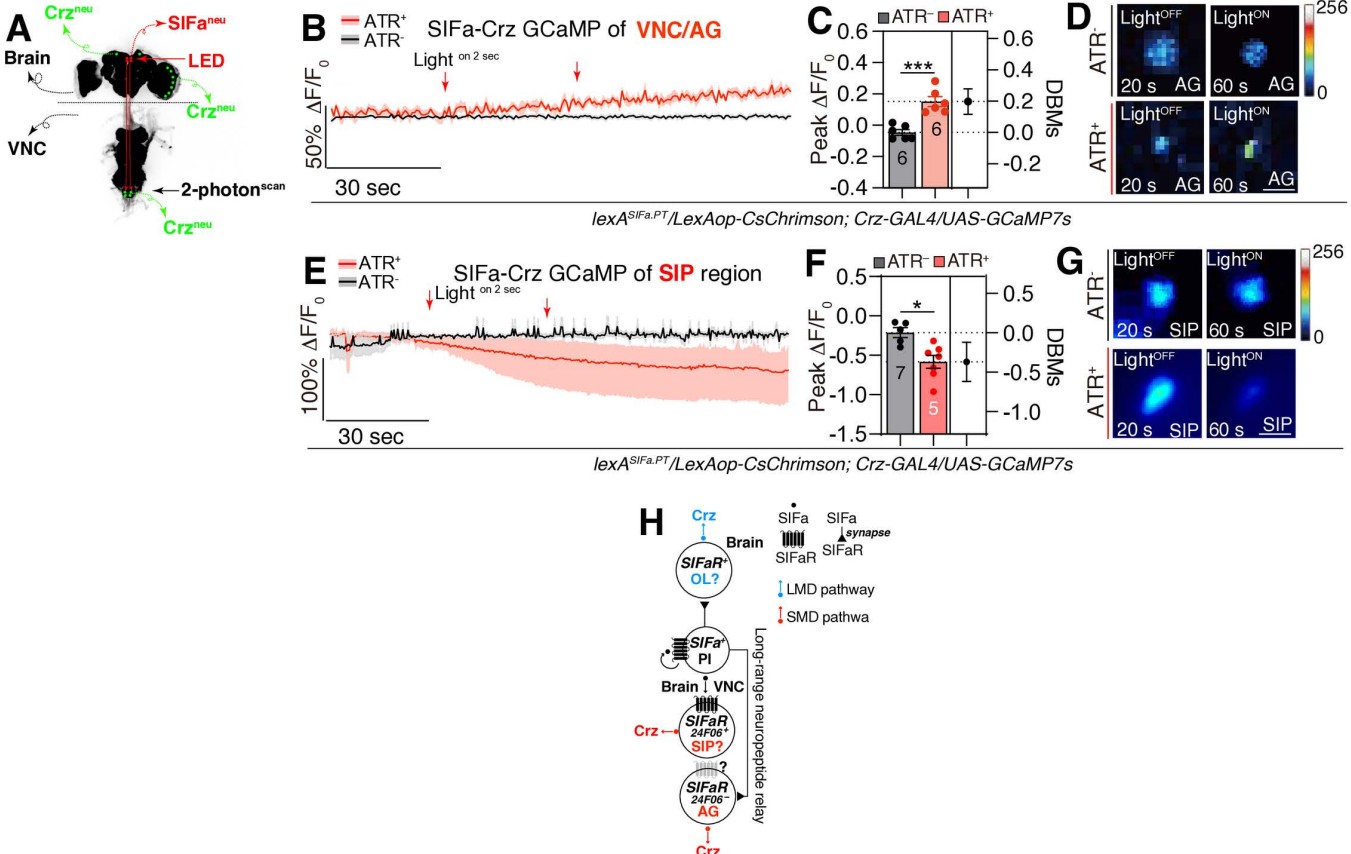

**Fig 7. *SIFa* neuron activation differentially modulates *Crz*+ neurons in AG and SIP. (A)** Schematic diagram depicting the precise location of the two-photon scanning procedure. Green labels indicate the nucleus of *Crz-GAL4*-labeled neurons located in AG, SIP, and OL region. Red labels indicate the nucleus of SIFa neurons located in PI region. The fly brain and VNC were immunostained by anti-nc82. **(B–D)** Crzs respond to SIFas activity in AG region. Fluorescence changes ($\Delta F/F_0$) of GCaMP7s in Crzs after optogenetic stimulation of SIFas (two-tailed unpaired *t* test). In all plots and statistical tests. Data are presented as mean ± s.e.m. ns = not significant ($p > 0.05$), *$p < 0.05$, **$p < 0.01$, ***$p < 0.001$, ****$p < 0.0001$. Sample sizes (*n*) are indicated in the figure panels. LED light was fired 2 s after 30 s of dark. *N* = 3 in each group. Scale bars represent 10 μm. **(E–G)** Crzs are negatively regulate by SIFas in SIP. Fluorescence changes ($\Delta F/F_0$) of GCaMP7s in Crzs after optogenetic stimulation of SIFas (two-tailed unpaired *t* test). In all plots and statistical tests. Data are presented as mean ± s.e.m. ns = not significant ($p > 0.05$), *$p < 0.05$, **$p < 0.01$, ***$p < 0.001$, ****$p < 0.0001$. Sample sizes (*n*) are indicated in the figure panels. LED light was fired 2 s after 30 s of dark. *N* = 8 to 5 in each group. Scale bars represent 10 μm. **(H)** Illustration depicting the mechanism by which Crz modulates LMD and SMD via SIFa-SIFaR signaling pathway. Underlying data for all graphs can be found in file S1 Data.

(Figs 7B–7D, S7G–S7I), evidenced by a sustained activity in intracellular Ca²⁺ levels that persisted in a high level compared to control ATR⁻ condition which shows gradual declining to baseline levels (Fig 7B–7D). In contrast to the OL and AG regions, the cells in the upper region of the SIP consistently show a decrease in Ca²⁺ levels following stimulation of the SIFa neurons (Fig 7E–7G). These calcium level changes were in contrast to the control group (without ATR) (Figs 7B–7G; S7G–S7I). These findings confirm that Crz neurons in OL and AG are activated in response to SIFa neuronal activity, but the activity of Crz neurons in SIP is inhibited by the activation of SIFa neuron, reinforcing their role as postsynaptic effectors in the neural circuitry governed by SIFa neurons. Moreover, these results support a model where long-range neuropeptide signaling - specifically SIFa-SIFaR in the brain coupled with Crz-CrzR in the AG—contributes significantly to the neural mechanisms governing interval timing (Fig 7H).

The majority of CrzR expression can be found in oenocytes (S8A Fig), and adult oenocytes (S8B Fig) have an enrichment of CrzR, according to the analysis of the fly Scope scRNA sequencing dataset [83]. However, the suppression of CrzR in oenocytes did not have any impact on both LMD and SMD behaviors in adults (S8C and S8D Fig) compared to controls (S8J and S8K Fig), indicating that oenocyte-expressing CrzR is not necessary for interval timing behaviors. Remarkably, the pan-neuronal suppression of CrzR exhibits no discernible impact on either LMD or SMD (S8E and S8F Fig) compared to controls (S8L and S8M Fig). Interestingly, our findings indicate that the downregulation of CrzR in the glial population resulted in the disruption of LMD rather than SMD (S8G and S8H Fig) compared to controls (S8N and S8O Fig). The efficiency of the CrzR-RNAi line has been confirmed by quantitative PCR (S8I Fig). All these data indicate that the relay of SIFa-SIFaR/Crz-CrzR neuropeptide signaling necessitates the presence of non-neuronal population in order to regulate interval timing behaviors.

## Discussion

Here, we investigated into the mechanisms by which the neuropeptide SIFa and its receptor SIFaR regulate two distinct interval timing behaviors in male *Drosophila melanogaster*: Longer-Mating-Duration (LMD) and Shorter-Mating-Duration (SMD). Utilizing RNAi and genetic rescue techniques, we identified that the expression of SIFaR in specific neuronal populations is crucial for sustaining LMD and SMD (Figs 1 and 2). We further explored the synaptic connections between SIFa and SIFaR neurons, uncovering that social context and sexual experience can lead to synaptic reorganization, thereby influencing the internal states of the CNS (Figs 3–5). Our study provides evidence that neuropeptide relay pathways involving SIFa-SIFaR/Crz-CrzR are integral in generating these distinct interval timing behaviors. Furthermore, the activity of Crz neurons, which is influenced by SIFa neurons, contributes to the neuronal mechanisms we refer to as neuropeptide relay, which are fundamental to the measurement of interval timing (Figs 6 and 7). These insights contribute to a broader understanding of the neural circuitry that underlies interval timing, with potential implications for a range of behavioral adaptations modulated by neuropeptidergic system.

Neuropeptides encompass a broad and heterogeneous group of signaling molecules that are synthesized by various cellular entities. As neuromodulators, co-transmitters, or circulating hormones, various neuropeptides have pleiotropic effects. However, there is a lack of understanding of the mechanisms by which neuropeptides can exhibit pleiotropic roles in context-dependent behaviors. The fact that numerous neuropeptides exhibit pleiotropic effects does not prevent us from gaining a comprehensive grasp of their functions at the organismal level. The autonomous functioning of peptidergic neurons is infrequent, as they mostly incorporate the modulatory influences of other peptidergic systems. Nevertheless, the process of achieving balance in this integration remains poorly understood [3,82].

Knockdown of *SIFaR* in *GAL4^24F06^*-expressing neurons induces a disinhibition-like phenomenon. Disinhibition, characterized by the alleviation of inhibitory constraints, permits the activation of neural circuits that are ordinarily repressed. This process is instrumental in sculpting behavioral patterns and facilitating the sequential progression of behaviors. Through the orchestrated promotion of select neuronal activation and concurrent inhibition of competing neural routes, disinhibition empowers the brain with the ability to dynamically ascertain and preserve the requisite behavioral state, concurrently smoothing the transition to ensuing behavioral phases [84]. It is known that *Drosophila* neural circuits also exhibit disinhibition phenotypes in light preference and ethanol sensitization [66,67]. Further investigation is needed to uncover the underlying mechanisms of this disinhibition-like phenotype observed in LMD and SMD behaviors.

SIFa-producing neurons are composed of a distinct quartet of neurons that exhibit extensive arborization across the majority of brain regions and AG. Therefore, the SIFa system is regarded as a typical neuropeptidergic system due to its ability to incorporate internal signals that are dependent on the state or context into the orchestrating peptidergic system. SIFa neurons integrate multiple inputs, which may contain encoded external and internal cues, as well as fundamental conditions like metabolic status, across numerous circuits. These particular large peptidergic neurons have the capability

to function as components of circuits that assess internal states and sensory inputs, thereby establishing transitions between conflicting behaviors such as mating and aggression or feeding and sleep [4,7,48].

An obstacle in understanding the complex role of neuropeptides is the scarcity of information regarding the cellular arrangement of neuropeptide receptors, which directly influence neuromodulatory circuits. Except for a few cases, we lack knowledge regarding the precise location of the receptor protein. In this study, we demonstrated the ability of a certain subset of neurons expressing SIFaR to selectively regulate two unique interval timing behaviors. Additionally, our findings demonstrate that the neuropeptide relay mediated by SIFa-SIFaR/Crz-CrzR has the potential to serve as a viable model for modulating context-dependent behavior through alterations in synaptic plasticity and calcium activity. Additionally, our study demonstrated that CrzR-expressing cells in non-neuronal populations play a distinct role in interval timing behaviors. It is important to consider that the 50% knockdown of SIFaR and CrzR may be sufficient to disrupt LMD and/or SMD behavior. However, the lack of phenotype with *repo-GAL4* or *elav^c155* could be due to a less efficient knockdown. This possibility highlights the need for cautious interpretation of negative behavioral data. Although our study reveals a key role for the SIFa-SIFaR pathway in regulating interval timing, the *GAL4^24F06*-labeled neuronal population likely integrates additional neuromodulatory and sensory inputs. The observed behavioral effects may thus reflect a convergence of signals beyond SIFa alone. Future studies combining single-cell transcriptomics to resolve molecular heterogeneity with subpopulation-specific functional manipulations (e.g., intersectional genetics or input mapping) could delineate how distinct circuit elements process these diverse inputs to collectively govern temporal control. Such approaches would clarify whether functional subdivisions exist within this population and how they contribute to behavioral flexibility across contexts.

Employing two distinct yet comparable models of interval timing behavior, LMD and SMD, we demonstrated that differential SIFa to SIFaR signaling is capable of modulating context-dependent behavioral responses. Synaptic strengths between SIFa and SIFaR neurons were notably enhanced in group-reared naïve males. However, these synaptic strengths specifically diminished in the OL, CB, and AG when males were singly reared, with a particular decrease in the AG region when males were sexually experienced (Fig 4A–4G). Intriguingly, overall calcium signaling within SIFaR^24F06 neurons was significantly reduced in group-reared naïve males, yet these signals surged dramatically in the OL with social isolation and in the AG with sexual experience (Fig 4I–4O). These calcium signals, as reported by the transcriptional calcium reporter CaLexA, were corroborated by GCaMP live imaging in both the AG and OL regions (Figs 5J–5L, 7B–7D, and S7G–S7I), indicating a close association between elevated calcium levels and LMD and SMD behaviors. The modulation of context-dependent synaptic plasticity and calcium dynamics by the SIFa neuropeptide through a single SIFaR receptor raises the question of how a single receptor can elicit such diverse responses. Recent neuroscientific studies in *Drosophila* have shown that individual neurons can produce multiple neurotransmitters and that neuropeptides are often colocalized with small molecule neurotransmitters [61,85–87]. Consistent with this, we have previously reported that SIFa neurons utilize a variety of neurotransmitters, including glutamate and dopamine [48]. Therefore, we propose that the SIFa-SIFaR-Crz-CrzR neuropeptidergic relay circuitry may interact with different neurotransmitters in distinct neuronal subpopulations to regulate context-dependent behaviors. Supporting this hypothesis, glutamate, known to function as an inhibitory neurotransmitter in the olfactory pathway of *Drosophila* [88], may be one such candidate. We speculate that neuropeptide cotransmission could underlie the mechanisms facilitating these complex, context-dependent behavioral patterns. Further research is warranted to elucidate how such cotransmission contributes to the intricate behavioral repertoire of the fly.

The relationship between SIFaR and gonadotropin inhibitory hormone receptors (GnIHR) [60] highlights an evolutionary connection, suggesting that both receptor types may have originated from a common ancestral precursor [89,90]. This study extends the foundational work of Martelli and colleagues, demonstrating that SIFamide not only regulates homeostatic behaviors such as feeding but also plays a crucial role in reproductive behavior [91]. GnIHR regulates food intake and reproductive behavior in a contrasting manner, thus prioritizing feeding behaviors during times of increased metabolic demand [92]. The evolutionary path of these behavioral regulatory mechanisms suggests a complex network of neuropeptide interactions that concurrently regulate physiological states and reproductive strategies. SIFamide influences various behaviors, such as feeding

and sexual activity, making it a crucial element in understanding how organisms adjust their reproductive strategies in relation to environmental and internal signals. The synthesis of these behavioral regulatory pathways underscores the evolutionary significance of SIFamide signaling in coordinating essential life-sustaining functions in *Drosophila* melanogaster and possibly in other taxa, thereby clarifying the mechanisms through which neuropeptides influence behavior in various contexts.

The presence of multiple different neurons that regulate the duration of mating in AG has been reported. According to Taylor and colleagues [80], the use of KCNJ2 to block fru-positive Crz-expressing AG neurons results in an increase in MD up to 100 min. Our findings demonstrate that cells expressing SIFaR in AG play a crucial role in regulating LMD and SMD behaviors. Due to the absence of fru- or dsx-positive neurons in SIFaR²⁴ᶠ⁰⁶ cells, we classify these cell types as separate from previously documented Crz neurons. Notably, we observed that SMD behavior is unaffected in both scenarios when Crz neurons significantly prolong the MD (Fig 6L and 6M). This observation suggests that the modulation of MD by Crz can still be influenced by neuropeptide relay signals from the PI region of SIFa to AG of SIFaR. This finding further demonstrates that the transmission of neuropeptides through SIFa can indeed induce behavior that is context-dependent, irrespective of the downstream neuronal circuits involved.

Neuropeptide relays are fundamental for survival and goal-oriented behaviors. One apparent case is the balance between appetite and satiety via orexigenic and anorexigenic neuronal circuits. It has been known that neuropeptides and hormones are important neural substrates for modulating appetite and satiety control [93,94]. SIFa has also been known as a central modulator in parallel inhibition between orexigenic and anorexigenic pathways [41]. Further investigations on SIFa-mediated neuropeptide relay will provide a foundational understanding of how neuropeptides and their receptors modulate neuronal circuits.

## Materials and methods

### Fly stocks and husbandry

*Drosophila melanogaster* were raised on cornmeal-yeast medium at similar densities to yield adults with similar body sizes. Flies were kept in 12 h light: 12 h dark cycles (LD) at 25 °C (ZT 0 is the beginning of the light phase, ZT12 beginning of the dark phase) except for some experimental manipulation. For temperature-controlled experiments, including those utilizing the temperature-sensitive tub-GAL80ᵗˢ driver, the flies were initially crossed and maintained at a constant temperature of 22 °C within an incubator. The temperature shift was initiated post-eclosion. Once the flies had emerged, they were transferred to an incubator set at an elevated temperature of 29 °C for a defined period, after which the experimental protocols were carried out. Wild-type flies were *Canton-S* (*CS*).

Following lines used in this study, *Canton-S* (#64349), *Df(1)Exel6234* (#7708), *Crz-GAL4* (#51977), *elavᶜ¹⁵⁵; UAS-Dicer* (#25750), *lexAop-CD8GFP; UAS-mLexA-VP16-NFAT, lexAop-rCD2-GFP* (#66542), *lexAop-nSyb-spGFP1-10, UAS-CD4-spGFP11* (#64315), *UAS-post-t-GRASP, lexAop2-pre-t-GRASP* (#79039), *UAS-pre-t-GRASP, lexAop2-post-t-GRASP* (#79040), *PromE-GAL4* (#65405), *repo-GAL4* (#7415), *SIFaR^B322* (#16202), *GAL4²³ᴳ⁰⁶* (#49041), *GAL4²⁴ᴬ¹²* (#49061), *GAL4²⁴ᶠ¹⁰* (#49087), *GAL4⁵⁷ᶠ¹⁰* (#46391), *lexA²⁴ᶠ⁰⁶* (#52695), *SIFaR-RNAi^JF01849* (#25831), *SIFaR-RNAi^HMS00299* (#34947), *tub-GAL80ᵗˢ* (#7108), *UAS-CD4tdGFP* (#35839), *UAS-RedStinger* (#8546), *UAS-FRT-stop-FRT-myrGFP* (#55811), *fru-FLP* (#66870), *CrzR-RNAi^JF02042* (#26017), *lexAop-flp* (#90892), *mCherry-RNAi* (#35785), *UAS-Denmark, UAS-syt.eGFP* (#33065), *UAS-KCNJ2* (#6596), *UAS-mCD8RFP, lexAop-mCD8GFP* (#32229), *UAS-GCaMP7s* (#80905), *AstA-GAL4^T2A* (#84593), *FMRFa-GAL4^T2A* (#84633), *Proc-GAL4* (#51972), and *lexAop-CsChrimson* (#55138) were obtained from the Bloomington *Drosophila* Stock Center at Indiana University. The following lines, *SIFaR-RNAi^v1783*(#1783), and *Crz-RNAi^v30670* (#30670) were obtained from the Vienna *Drosophila* Resource Center. The following lines, *SIFa-lexA^T2A* (#FBA00116), *SIFa-GAL4^T2A* (#FBA00103), *SIFaR-GAL4^T2A* (#FBF00102), *Capa-GAL4^T2A* (#FB100010), *Ms-GAL4^T2A* (#FBA00065), *Lk-GAL4^T2A* (#FBA00235), and *SIFaR-lexA^T2A* (#FBF00086) were obtained from Qidong Fungene Biotechnology in China. The *GAL4^SIFa.PT* was a gift from Jan A. Veenstra. The *UAS-SIFaR* was a gift from Young-Joon Kim in GIST. The fly genotypes used in each figure of this study are listed in Table 2.

**Table 2. Genotypes of flies used for experiments in this study.**

| Figure panel | Genotype |
|---|---|
| Fig 1A and 1B | elav^c155; SIFaR-RNAi |
| Fig 1C and 1D | SIFaR-RNAi/+ |
| Fig 1E and 1F | elav^c155/+ |
| Fig 1G and 1H | elav^c155; SIFaR-RNAi, elav-GAL80 |
| Fig 1I and 1J | elav-GAL80, elav^c155/+ |
| Fig 1K | tub-GAL4; SIFaR-RNAi, tub-GAL80^ts and +/SIFaR-RNAi |
| Fig 1L | SIFaR^2A; UAS-CD4tdGFP, UAS-RedStinger |
| Fig 2B and 2C | GAL4^24F06; SIFaR-RNAi, UAS-dicer |
| Fig 2D and 2E | SIFaR-RNAi, UAS-dicer/+ |
| Fig 2F and 2G | GAL4^24F06/UAS-SIFaR; SIFaR^B322/SIFaR^B322 |
| Fig 2H and 2I | GAL4^24F06/+ |
| Fig 2J and 2K | UAS-SIFaR/+ |
| Fig 2L | GAL4^24F06; UAS-mCD8GFP |
| Fig 2M | GAL4^24F06; UAS-RedStinger |
| Fig 3A, 3C–3J | SIFa^2A-lexA, GAL4^24F06; lexAop-nSyb-spGFP1–10, UAS-CD4-spGFP11 |
| Fig 4A–4G | GAL4^SIFa.PT, lexA^SIFaR-2A; lexAop-2-post-t-GRASP, UAS-pre-t-GRASP |
| Fig 4I–4O | GAL4^24F06; lexAop-CD2-GFP, UAS-mLexA-VP16-NFAT, lexAop-CD8-GFP-A2-CD8-GFP |
| Fig 5A–5G | GAL4^24F06; UAS-syt.eGFP |
| Fig 5H–5L | lexA^SIFa.PT/lexAop-CsChrimson; GAL4^24F06/UAS-GCaMP7s |
| Fig 6A | Crz-GAL4, lexA^24F06; UAS-mCD8RFP, lexAop-mCD8GFP |
| Fig 6B and 6C | Crz-GAL4, lexA^24F06; UAS-Stinger, lexAop-tdTomato.nls |
| Fig 6D and 6E | GAL4^24F06; Crz-RNAi |
| Fig 6F and 6G | Crz-RNAi/+ |
| Fig 6H and 6I | Crz-GAL4; SIFaR-RNAi |
| Fig 6J and 6K | Crz-GAL4/+ |
| Fig 6L and 6M | Crz-GAL4; UAS-KCNJ2 |
| Fig 6N and 6O | UAS-KCNJ2/+ |
| Fig 7B–7G | lexA^SIFa.PT/lexAop-CsChrimson; Crz-GAL4/UAS-GCaMP7s |
| Table 1 | See Table |
| S1A and S1B Fig | Canton-S |
| S1C and S1D Fig | repo-GAL4; SIFaR-RNAi |
| S1E and S1F Fig | elav^c155; SIFaR-RNAi (v1783) |
| S1G and S1H Fig | elav^c155; SIFaR-RNAi (JF01849) |
| S1I and S1J Fig | elav^c155; SIFaR-RNAi (HMS00299) |
| S1K–S1N Fig | elav^c155; SIFaR-RNAi, tub-GAL80^ts |
| S1O and S1P Fig | tub-GAL4; SIFaR-RNAi (HMS00299) |
| S1Q and S1R Fig | nsyb-GAL4; SIFaR-RNAi (HMS00299) |
| S2A and S2B Fig | GAL4^23G06; SIFaR-RNAi, UAS-dicer |
| S2C and S2D Fig | GAL4^24A12; SIFaR-RNAi, UAS-dicer |
| S2E and S2F Fig | GAL4^57F10; SIFaR-RNAi, UAS-dicer |
| S2G Fig | lexA^SIFaR.2A, GAL4^24F06; UAS-mCD8RFP, lexAop-mCD8GFP |
| S2H Fig | GAL4^24F06, lexA^SIFaR.2A; UAS-stop-myrGFP, lexAop-flp |

*(Continued)*

**Table 2.** (Continued)

| Figure panel | Genotype |
|---|---|
| S3A–S3E Fig | $GAL4^{SIFa.PT}$, $lexA^{24F06}$; UAS-mCD8RFP, lexAop-mCD8GFP |
| S4C–S4L Fig | $SIFa^{2A-lexA}$, $GAL4^{24F06}$; lexAop-nSyb-spGFP1–10, UAS-CD4-spGFP11 |
| S5A–S5E Fig | $GAL4^{SIFa.PT}$, $lexA^{SIFaR-2A}$; lexAop-2-post-t-GRASP, UAS-pre-t-GRASP |
| S5F–S5L Fig | $GAL4^{24F06}$; lexAop-CD2-GFP, UAS-mLexA-VP16-NFAT, lexAop-CD8-GFP-A2-CD8-GFP |
| S6A–S6I Fig | $lexA^{SIFaR-2A}$, $GAL4^{SIFa.PT}$; lexAop-2-pre-t-GRASP, UAS-post-t-GRASP |
| S7A–S7D Fig | $GAL4^{24F06}$; UAS-syt.eGFP |
| S7E Fig | Crz-GAL4, $lexA^{24F06}$; UAS-mCD8RFP, lexAop-mCD8GFP |
| S7F Fig | Crz-GAL4, $lexA^{24F06}$; UAS-Stinger, lexAop-tdTomato.nls |
| S7G–S7I Fig | $lexA^{SIFa.PT}$/lexAop-CsChrimson; Crz-GAL4/UAS-GCaMP7s |
| S8C and S8D Fig | PromE-GAL4, tub-GAL80$^{ts}$; CrzR-RNAi |
| S8E and S8F Fig | $elav^{c155}$; CrzR-RNAi |
| S8G and S8H Fig | repo-GAL4; CrzR-RNAi |
| S8I Fig | tub-GAL4; CrzR-RNAi, tub-GAL80$^{ts}$ and +/CrzR-RNAi |
| S8J and S8K Fig | PromE-GAL4, tub-GAL80$^{ts}$/+ |
| S8L and S8M Fig | CrzR-RNAi/+ |
| S8N and S8O Fig | repo-GAL4/+ |
| S1 Table | See Table |
| S2 Table | See Table |
| S1 Supplemental information | See figure |
| S2 Supplemental information | SIFa-PT-to-SIFaR-2A-tGRASP raw data |
| S3 Supplemental information | $GAL4^{24F06}$-CalexA raw data |

The CS background was selected as the experimental background due to its well-characterized and consistent LMD and SMD behaviors. To ensure that genetic variation did not confound our results, all GAL4, UAS, and RNAi lines employed in our assays were rigorously backcrossed into the CS strain, often exceeding ten generations of backcrossing. This approach was undertaken to isolate the effects of our genetic manipulations from those of genetic background. We assert that the extensive backcrossing to the CS background, in concert with the internal control in LMD and SMD, provides a stable platform for the accurate interpretation of the LMD and SMD phenotypes observed in our experiments. To reduce the variation from genetic background, all flies were backcrossed for at least 10 generations to CS strain. For the generation of outcrosses, all GAL4, UAS, and RNAi lines employed as the virgin female stock were backcrossed to the CS genetic background for a minimum of ten generations. Notably, the majority of these lines, which were utilized for LMD assays, have been maintained in a CS backcrossed state for long-term generations subsequent to the initial outcrossing process, exceeding ten backcrosses. Based on our experimental observations, the genetic background of primary significance is that of the X chromosome inherited from the female parent. Consequently, we consistently utilized these fully outcrossed females as virgins for the execution of experiments pertaining to LMD and SMD behaviors. Contrary to the influence on LMD behaviors, we have previously demonstrated that the genetic background exerts negligible influence on SMD behaviors, as reported in our prior publication [36]. The mutants and transgenic lines utilized in this study have been previously characterized, with the exception of the novel transgenic strains that we generated and describe herein.

## Mating duration assay

The MD assay in this study has been reported [34–36]. To enhance the efficiency of the MD assay, we utilized the *Df(1) Exel6234* (DF hereafter) genetic modified fly line in this study, which harbors a deletion of a specific genomic region that includes the sex peptide receptor (SPR) [95,96]. Previous studies have demonstrated that virgin females of this line exhibit increased receptivity to males [96]. We conducted a comparative analysis between the virgin females of this line and the CS virgin females and found that both groups induced SMD. Consequently, we have elected to employ virgin females from this modified line in all subsequent studies. For group-reared (naïve) males, 40 males from the same strain were placed into a vial with food for 5 days. For single-reared males, males of the same strain were collected individually and placed into vials with food for 5 days. For experienced males, 40 males from the same strain were placed into a vial with food for 4 days then 80 DF virgin females were introduced into vials for last 1 day before assay. Forty DF virgin females were collected from bottles and placed into a vial for 5 days. These females provide both sexually experienced partners and mating partners for MD assays. At the fifth day after eclosion, males of the appropriate strain and DF virgin females were mildly anaesthetized by $CO_2$. After placing a single female in to the mating chamber, we inserted a transparent film then placed a single male to the other side of the film in each chamber. After allowing for 1 h of recovery in the mating chamber in 25 °C incubators, we removed the transparent film and recorded the mating activities. Only those males that succeeded to mate within 1 h were included for analyses. Initiation and completion of copulation were recorded with an accuracy of 10 s, and total MD was calculated for each couple. Part of genetic controls with *GAL4*/+ or *UAS*/+ lines were omitted from supplementary figures, as prior data confirm their consistent exhibition of normal LMD and SMD behaviors [34–37,59]. Hence, genetic controls for LMD and SMD behaviors were incorporated exclusively when assessing novel fly strains that had not previously been examined. In essence, internal controls were predominantly employed in the experiments, as LMD and SMD behaviors exhibit enhanced statistical significance when internally controlled. Within the LMD assay, both group and single conditions function reciprocally as internal controls. A significant distinction between the naïve and single conditions implies that the experimental manipulation does not affect LMD. Conversely, the lack of a significant discrepancy suggests that the manipulation does influence LMD. In the context of SMD experiments, the naïve condition (equivalent to the group condition in the LMD assay) and sexually experienced males act as mutual internal controls for one another. A statistically significant divergence between naïve and experienced males indicates that the experimental procedure does not alter SMD. Conversely, the absence of a statistically significant difference suggests that the manipulation does impact SMD. Hence, we incorporated supplementary genetic control experiments solely if they deemed indispensable for testing. All assays were performed from noon to 4 PM. We conducted blinded studies for every test.

## Generation of transgenic flies

To generate the *SIFa^PT^-lexA* driver, the putative promotor sequence of the gene was amplified by PCR using wild-type genomic DNA as a template with the following primers GCCAATTGGCTGAATCTCCTGACCCTCA and GCAGATCTCTTGCAGTTTTCGGTGAGC as mentioned before [78]. The amplified DNA fragment (1482 base pairs located immediately upstream of the *SIFa* coding sequence) was inserted into the E2 Enhancer-lexA vector. This vector, supplied by Qidong Fungene Biotechnology Co., Ltd. (http://www.fungene.tech/), is a derivative of the pBPLexA::p65Uw vector (available at https://www.addgene.org/26231). The insertion was achieved by digesting the fragment and the vector with EcoRI and XbaI restriction enzymes to create compatible sticky ends. The genetic construct was inserted into the *attp2* site on chromosome III to generate transgenic flies using established techniques, a service conducted by Qidong Fungene Biotechnology Co., Ltd.

## Immunostaining

The dissection and immunostaining protocols for the experiments are described elsewhere [34]. After 5 days of eclosion, the *Drosophila* brain was taken from adult flies and fixed in 4% formaldehyde at room temperature for 30 min. The sample

was washed three times (5 min each) in 1% PBT and then blocked in 5% normal goat serum for 30 min. The sample next be incubated overnight at 4 °C with primary antibodies in 1% PBT, followed by the addition of fluorophore-conjugated secondary antibodies for one hour at room temperature. The brain was mounted on plates with an antifade mounting solution (Solarbio) for imaging purposes.

Samples were imaged with Zeiss LSM880. Antibodies were used at the following dilutions: Chicken anti-GFP (1:500, Invitrogen), mouse anti-nc82 (1:50, DSHB), rabbit anti-DsRed (1:500, Rockland Immunochemicals), Alexa-488 donkey anti-chicken (1:200, Jackson ImmunoResearch), Alexa-555 goat anti-rabbit (1:200, Invitrogen), Alexa-647 goat anti-mouse (1:200, Jackson ImmunoResearch).

## Quantitative analysis of fluorescence intensity

To ascertain calcium levels and synaptic intensity from microscopic images, we dissected and imaged five-day-old flies of various social conditions and genotypes under uniform conditions. For group-reared (naïve) flies, the flies were reared in group condition and dissect right after 5 days of rearing without any further action. For single-reared flies, the flies were reared in single condition and dissect at the same time as group-reared flies right after 5 days of rearing without any further action. For sexual experienced flies, the flies were reared in group condition after 4 days of rearing and will be given virgins to give them sexual experience for 1 day, those flies will also be dissected at the same time as group and single-reared flies after one day. The GFP signal in the brains and VNCs was amplified through immunostaining with chicken anti-GFP primary antibody. Image analysis was conducted using ImageJ software. For the quantification of fluorescence intensities, an investigator, blinded to the fly's genotype, thresholded the sum of all pixel intensities within a sub-stack to optimize the signal-to-noise ratio, following established methods [70]. The total fluorescent area or region of interest (ROI) was then quantified using ImageJ, as previously reported. For CaLexA signal quantification, we adhered to protocols detailed by Kayser and colleagues [97], which involve measuring the ROI's GFP-labeled area by summing pixel values across the image stack. This method assumes that changes in the GFP-labeled area are indicative of alterations in the CaLexA signal, reflecting synaptic activity. ROI intensities were background-corrected by measuring and subtracting the fluorescent intensity from a non-specific adjacent area, as per Kayser and colleagues [97]. For the analysis of GRASP or tGRASP signals, a sub-stack encompassing all synaptic puncta was thresholded by a genotype-blinded investigator to achieve the optimal signal-to-noise ratio. The fluorescence area or ROI for each region was quantified using ImageJ, employing a similar approach to that used for CaLexA quantification [70].

## Image display and thresholding for figure preparation

To prepare representative images for figures, grayscale channels were used for single-fluorophore display to improve clarity and avoid color-based misinterpretation. For overlays involving multiple reporters (e.g., GFP, RFP, and nc82), distinct non-overlapping pseudocolors were applied, with channel identities labeled below each panel. Thresholding for all displayed images was performed conservatively and consistently across genotypes and conditions, with the primary goal of preserving visible biological signals while avoiding oversaturation. For quantified data, identical threshold settings were applied across conditions. All brightness and contrast adjustments were applied uniformly to the entire image and were restricted to linear modifications for visual clarity only. No nonlinear enhancements or selective adjustments were applied. These standardized procedures ensured both accurate visualization and reproducible quantification. Raw image data are available upon request.

## Colocalization analysis

Before the colocalization analysis, an investigator, blinded to the fly's genotype, thresholded the sum of all pixel intensities within a sub-stack to optimize the signal-to-noise ratio, following established methods [70]. To perform colocalization analysis of multi-color fluorescence microscopy images in this study, we employed ImageJ software [98]. In brief, we merged

image channels to form a composite with accurate color representation and applied a threshold to isolate yellow pixels, signifying colocalization. The measured "area" values represented the colocalization zones between fluorophores. To determine the colocalization percentage relative to the total area of interest (e.g., GFP or RFP), we adjusted thresholds to capture the full fluorophore areas and remeasured to obtain total areas. The quantification of the overlap was performed using confocal images with projection by standard deviation function provided by ImageJ to ensure precise measurements and avoid pixel saturation artifacts. The colocalization efficiency was calculated by dividing the colocalized area by the total fluorophore area. All samples were imaged uniformly.

### Particle analysis

Before the particle analysis, an investigator, blinded to the fly's genotype, thresholded the sum of all pixel intensities within a sub-stack to optimize the signal-to-noise ratio, following established methods [70]. To quantitatively measure particle intensity of cell number and synaptic puncta in microscopic images, we applied ImageJ software. Initially, the image is converted to grayscale to reduce complexity and enhance contrast. Subsequently, thresholding techniques are employed to binarize the image, distinguishing particles from the background. This binarization can be achieved through automated thresholding algorithms or manual adjustment to optimize the segmentation. The results of these measurements are then available for review by conducting "Analyze Particles" function of ImageJ. All specimens were imaged under identical conditions.

### Quantitative RT-PCR

The expression levels of SIFaR and CrzR after knockdown by *SIFaR-RNAi* lines and *CrzR-RNAi* were analyzed by quantitative real-time RT-PCR with SYBR Green qPCR MasterMix kit (Selleckchem). The primers of SIFaR are F:5′-AAGCAGGAGAGCGAGTTCAG-3′; R: 5′-TTCGCCTTGTTTTGTCACAG-3′ [99]. The primers of CrzR are F: 5′-TGGTG CCGCTCCTGAAATCGT-3′; R: 5′-TTCCGCCATCGGTGGTGCTTC-3′. qPCR reactions were performed in triplicate, and the specificity of each reaction was evaluated by dissociation curve analysis. Each experiment was replicated three times. PCR results were recorded as threshold cycle numbers (Ct). The fold change in the target gene expression, normalized to the expression of internal control gene (GAPDH) and relative to the expression at time point 0, was calculated using the $2^{-\Delta\Delta CT}$ method as previously described [100]. The results are presented as the mean $\pm$ SD of three independent experiments.

### Feeding of retinal

All trans-retinal powder (Sigma) was dissolved in EtOH as a 100 mM stock solution. 200 µL of this stock solution was diluted in 50 mL of melted normal food to prepare 400 µM of all trans-retinal (ATR) food. For CsChrimson experiments, male flies of proper genotype were selected and separated into two groups (with and without ATR) after 5 days of eclosion for at least 3 days prior to any optogenetic experiments.

### Two-photon calcium imaging

To measure the neuron activity in AG region, the flies were carefully sedated with ice and then placed in an inverted posture on a plastic plate coated with UV glue, which securely held flies in the middle. Whole fly was exposed to AHL for red light activation. The plate was filled with *Drosophila* Adult Hemolymph-Like Saline (AHLS) buffer [108 mM NaCl, 5 mM KCl, 4 mM NaHCO$_3$, 1 mM NaH$_2$PO$_4$, 15 mM ribose, 5 mM Hepes (pH 7.5), 300 mosM, CaCl$_2$ (2 mM), and MgCl$_2$ (8.2 mM)] to ensure the flies' neurons remained in an active state throughout the experiment [101,102]. Subsequently, the flies were dissected to specifically expose the VNC for detailed examination. Calcium imaging was performed using Zeiss LSM880 microscope with a 20× water immersion objective. Images were acquired at 2 frames per second at a resolution of 512 × 512 pixels. GCaMP7 slow (GCaMP7s) signals were recorded with an 880 nm laser and optogenetic stimulation

was achieved with a 590–595 nm 20 W red light pulses at 50–60 Hz (0.16 mW/mm$^2$), LED stimulation lasts 2 s. ROIs were manually selected from the cell body in AG area with ImageJ.

$\Delta F/F_0 = (F_t − F_0)/F_0 \times 100\%$ and $Peak\ \Delta F/F_0 = (Peak\ F_t − F_0)/F_0 \times 100\%$ were used to determine the fluorescence change, where Ft is the fluorescence at time point n and $F_0$ is the fluorescence from the average intensity of 10 frames fluorescence before optogenetic stimulation. To quantify neuronal activity in the brain region, the brains of male flies with the appropriate genotype were dissected and secured in a plastic dish. All subsequent steps were conducted as previously described.

## Single-nucleus RNA-sequencing analyses

The snRNAseq dataset analyzed in this paper is published in [83] and available at the Nextflow pipelines (VSN, https://github.com/vib-singlecell-nf), the availability of raw and processed datasets for users to explore, and the development of a crowd-annotation platform with voting, comments, and references through SCope (https://flycellatlas.org/scope), linked to an online analysis platform in ASAP (https://asap.epfl.ch/fca). For the generation of the tSNE plots, we utilized the Fly SCope website (https://scope.aertslab.org/#/FlyCellAtlas/*/welcome). Within the session interface, we selected the appropriate tissues and configured the parameters as follows: 'Log transform' enabled, 'CPM normalize' enabled, 'Expression-based plotting' enabled, 'Show labels' enabled, 'Dissociate viewers' enabled, and both 'Point size' and 'Point alpha level' set to maximum. For all tissues, we referred to the individual tissue sessions within the '10X Cross-tissue' RNAseq dataset. Each tSNE visualization depicts the coexpression patterns of genes, with each color corresponding to the genes listed on the left, right, and bottom of the plot. The tissue name, as referenced on the Fly SCope website is indicated in the upper left corner of the tSNE plot. Dashed lines denote the significant overlap of cell populations annotated by the respective genes. Coexpression between genes or annotated tissues is visually represented by differentially colored cell populations. For instance, yellow cells indicate the coexpression of a gene (or annotated tissue) with red color and another gene (or annotated tissue) with green color. Cyan cells signify coexpression between green and blue, purple cells for red and blue, and white cells for the coexpression of all three colors (red, green, and blue). Consistency in the tSNE plot visualization is preserved across all figures.

Single-cell RNA-sequencing (scRNA-seq) data from the *Drosophila melanogaster* were obtained from the Fly Cell Atlas (FCA) website (https://doi.org/10.1126/science.abk2432). Oenocytes gene expression analysis employed UMI (Unique Molecular Identifier) data extracted from the 10× VSN oenocyte (Stringent) loom and h5ad file, encompassing a total of 506,660 cells. The Seurat (v4.2.2) package (https://doi.org/10.1016/j.cell.2021.04.048) was utilized for data analysis. Violin plots were generated using the "Vlnplot" function, the cell types are split by FCA.

## Connectome analysis

Whole brain connectomics data were obtained from VFB (https://v2.virtualflybrain.org) [76]. The right SIFa (VFB ID: VFB_jrchk542 and VFB_jrchk541) dataset was used to gather information on the synaptic connections between the presynaptic and the postsynaptic neurons of interest. The connectivity was visualized with Sankey diagram and doughnut diagram by the *NetworkD3* R package (https://github.com/christophergandrud/networkD3).

## Statistical tests

Statistical analysis of MD assay was described previously [34–36]. More than 50 males (naïve, experienced, and single) were used for MD assay. Our experience suggests that the relative MD differences between naïve and experienced condition and singly reared are always consistent; however, both absolute values and the magnitude of the difference in each strain can vary. So, we always include internal controls for each treatment as suggested by previous studies [103]. Therefore, statistical comparisons were made between groups that were naïvely reared, sexually experienced, and

singly reared within each experiment. As MD of males showed normal distribution (Kolmogorov–Smirnov tests, *p* > 0.05), we used two-sided Student's t tests. The mean ± standard error (s.e.m) (******=*p* < 0.0001, ***=*p* < 0.001, **=*p* < 0.01, *=*p* < 0.05). All analysis was done in GraphPad (Prism). Individual tests and significance are detailed in figure legends.

Besides traditional *t* test for statistical analysis, we added estimation statistics for all MD assays and two group comparing graphs. In short, 'estimation statistics' is a simple framework that—while avoiding the pitfalls of significance testing—uses familiar statistical concepts: means, mean differences, and error bars. More importantly, it focuses on the effect size of one's experiment/intervention, as opposed to significance testing [104]. In comparison to typical NHST plots, estimation graphics have the following five significant advantages such as (1) avoid false dichotomy, (2) display all observed values (3) visualize estimate precision (4) show mean difference distribution. And most importantly (5) by focusing attention on an effect size, the difference diagram encourages quantitative reasoning about the system under study [105]. Thus, we conducted a reanalysis of all our two group data sets using both standard *t* tests and estimate statistics. In 2019, the Society for Neuroscience journal eNeuro instituted a policy recommending the use of estimation graphics as the preferred method for data presentation [106].

## Supporting information

**S1 Fig. Screening of *SIFaR-RNAi* lines and quantification of *SIFaR²ᴬ* expression in CNS. (A, B)** LMD and SMD assays of *Conton-S* (WT) (two-tailed unpaired *t* test). In all plots and statistical tests. Data are presented as mean ± s.e.m. ns = not significant (*p* > 0.05), *p* < 0.05, **p* < 0.01, ***p* < 0.001, ****p* < 0.0001. Sample sizes (*n*) are indicated in the figure panels. **(C, D)** LMD and SMD assays for *repo-GAL4* mediated knockdown of SIFaR *via SIFaR-RNAi* (two-tailed unpaired *t* test). In all plots and statistical tests. Data are presented as mean ± s.e.m. ns = not significant (*p* > 0.05), *p* < 0.05, **p* < 0.01, ***p* < 0.001, ****p* < 0.0001. Sample sizes (*n*) are indicated in the figure panels. **(E–J)** LMD and SMD assays for *elav^c155*-mediated knockdown of SIFaR *via* (E, F) *SIFaR-RNAi (JF01849)*, (G, H) *SIFaR-RNAi (HMS00299)*, and (I, J) *SIFaR-RNAi (HMS00299)* (two-tailed unpaired *t* test). In all plots and statistical tests. Data are presented as mean ± s.e.m. ns = not significant (*p* > 0.05), *p* < 0.05, **p* < 0.01, ***p* < 0.001, ****p* < 0.0001. Sample sizes (*n*) are indicated in the figure panels. **(K, L)** LMD and SMD assays for *elav^c155*-mediated knockdown of SIFaR *via SIFaR-RNAi* together with *tub-GAL80^ts* in 29 °C (two-tailed unpaired *t* test). In all plots and statistical tests. Data are presented as mean ± s.e.m. ns = not significant (*p* > 0.05), *p* < 0.05, **p* < 0.01, ***p* < 0.001, ****p* < 0.0001. Sample sizes (*n*) are indicated in the figure panels. **(M, N)** LMD and SMD assays for *elav^c155*-mediated knockdown of SIFaR *via SIFaR-RNAi* together with *tub-GAL80^ts* in 18 °C (two-tailed unpaired *t* test). In all plots and statistical tests. Data are presented as mean ± s.e.m. ns = not significant (*p* > 0.05), *p* < 0.05, **p* < 0.01, ***p* < 0.001, ****p* < 0.0001. Sample sizes (*n*) are indicated in the figure panels. **(O–R)** *Drosophila* lethality induced by *nsyb-GAL4* and *tub-GAL4* knockdown of SIFaR *via SIFaR-RNAi.* Underlying data for all graphs can be found in file S1 Data.
(TIF)

**S2 Fig. Colocalization analysis between *GAL4²⁴ᶠ⁰⁶* and *lexA^SIFaR.2A*. (A–F)** LMD and SMD assays of (A, B) *GAL4²³ᴳ⁰⁶*, (C, D) *GAL4²⁴ᴬ¹²*, and (E, F) *GAL4⁵⁷ᶠ¹⁰* mediated knockdown of SIFaR *via SIFaR-RNAi* together with *UAS-dicer* (two-tailed unpaired *t* test). In all plots and statistical tests. Data are presented as mean ± s.e.m. ns = not significant (*p* > 0.05), *p* < 0.05, **p* < 0.01, ***p* < 0.001, ****p* < 0.0001. Sample sizes (*n*) are indicated in the figure panels. **(G)** Male flies brain and VNC expressing *lexA^SIFaR.2A* and *GAL4²⁴ᶠ⁰⁶* drivers together with *UAS-mCD8RFP* and *lexAop-mCD8GFP* were immunostained with anti-GFP (green), anti-DsRed (red), and anti-nc82 (blue) antibodies. Scale bars represent 100 μm. Boxes indicate the magnified regions of interest presented in the bottom panels. The panels presented as gray scale are to clearly show the axon projection patterns of neurons in brain and VNC labeled by *lexA^SIFaR.2A* and *GAL4²⁴ᶠ⁰⁶* driver. (H) Male flies brain and VNC expressing *lexA^SIFaR.2A* and *GAL4²⁴ᶠ⁰⁶* drivers together with *UAS-stop-myrGFP* and *lexAop-flp* were immunostained with anti-GFP (green) and anti-nc82 (blue). Underlying data for all graphs can be found in file S1 Data.
(TIF)

**S3 Fig.** *GAL4$^{SIFa.PT}$* and *lexA$^{24F06}$* **form strong synapses and *SIFa*-positive neurons connected to specific regions in CNS. (A)** Male flies brain expressing *GAL4$^{SIFa.PT}$* and *lexA$^{24F06}$* drivers together with *UAS-mCD8RFP* and *lexAop-mCD8GFP* were immunostained with anti-GFP (green), anti-DsRed (red), and anti-nc82 (blue) antibodies. Boxes and dashed circles indicate the magnified regions of interest presented in the bottom panels. The left two panels are presented as a gray scale to clearly show the axon projection patterns of neurons labeled by *GAL4$^{SIFa.PT}$* and *lexA$^{24F06}$* driver. Scale bars represent 100 μm in brain and VNC panels, and 25 μm in other panels. **(B–E)** Colocalization analysis of GFP and RFP staining, normalized to total GFP and RFP areas. Bars represent the mean GFP (green column) and RFP (red column) fluorescence level with error bars representing SEM. DBMs represent the difference between means. The regions analyzed are clearly marked on the figure. Asterisks represent significant differences, as revealed by the Student *t* test and ns represents non-significant difference (*$p < 0.05$, **$p < 0.01$, ***$p < 0.001$, ****$p < 0.0001$). The same symbols for statistical significance are used in all other figures. See the Materials and methods for a detailed description of the colocalization analysis used in this study. **(F)** Sankey diagram illustrating the connectivity of two SIFa neurons (PDM34) across brain right regions. The diagram visualizes synaptic connectivity data obtained from Virtual Fly Brain: "Connectivity per region for SIFa (FlyEM-HB:1418618235)" and "Connectivity per region for SIFa (PDM34) (FlyEM-HB:699031185)." Upstream regions (labeled "up-") represent brain areas projecting signals to SIFa neurons, while downstream regions (labeled "down-") represent brain areas receiving signals from these neurons. The thickness of each link corresponds to the number of synaptic connections. See the Materials and methods for a detailed description of the connectome analysis used in this study. **(G)** Sankey diagram illustrating the downstream connections of two SIFa neurons located in the right brain region to four specific regions: GA, AL, FB, and SMP. The contributions of SIFa neuron FlyEM-HB:1418618235 and FlyEM-HB:699031185 account for 22.3% and 30.2% of the total downstream synaptic connections, respectively. The thickness of each link corresponds to the number of synaptic connections in the selected regions. **(H)** Diagram of interconnection between *GAL4$^{SIFa.PT}$* and *lexA$^{24F06}$* neurons along with interconnected networks formed by *GAL4$^{24F06}$*. Underlying data for all graphs can be found in file S1 Data.
(TIF)

**S4 Fig.** *SIFa$^{2A-lexA}$* and *GAL4$^{24F06}$* **form synapses only in brain region. (A)** Virtual Fly Brain (VFB) visualization of SMP, GA, FB, and AL brain regions in *Drosophila*. In the last two panels, the gray area represents the overlapping region of SMP/FB/AL, and the red area represents GA. **(B)** Co-locolization between SIFa and AL in Virtual Fly Brain (VFB). White boxes indicate the magnified regions of interest presented in the right panel. The yellow area represents synaptic connections formed by *SIFa* (green) and AL (red). **(C)** GRASP assay for *SIFa$^{2A-lexA}$* and *GAL4$^{24F06}$* in AL region of naïve (top three columns), single (middle three columns) and experienced (bottom three columns) male flies. Male flies expressing *SIFa$^{2A-lexA}$*, *GAL4$^{24F06}$* and *lexAop-nsyb-spGFP1-10, UAS-CD4-spGFP11* were dissected after 5 days of growth. GFP is pseudo-colored as "Green fire blue". **(D, E)** Quantification of relative value for synaptic intensity (two-tailed unpaired *t* test). In all plots and statistical tests. Data are presented as mean ± s.e.m. ns = not significant ($p > 0.05$), *$p < 0.05$, **$p < 0.01$, ***$p < 0.001$, ****$p < 0.0001$. Sample sizes (*n*) are indicated in the figure panels. **(F)** The synaptic interactions visualized utilizing the GRASP system in naïve and single male flies. The GFP fluorescence was processed using ImageJ software, where a threshold function was applied to distinguish fluorescence from the background. **(G)** Quantification of synaptic puncta formed between *SIFa$^{2A-lexA}$* and *GAL4$^{24F06}$* in brain between naïve and single male flies. The synaptic interactions were visualized utilizing the GRASP system in male flies. Bars represent the mean particle number with error bars representing SEM. Asterisks represent significant differences, as revealed by the Student *t* test and ns represents non-significant difference (*$p < 0.05$, **$p < 0.01$, ***$p < 0.001$, ****$p < 0.0001$). Sample sizes (*n*) are indicated in the figure panels. See the Materials and methods for a detailed description of the particle analysis used in this study. **(H)** Quantification of average synapse size formed between *SIFa$^{2A-lexA}$* and *GAL4$^{24F06}$* in brain between naïve and single male flies (two-tailed unpaired *t* test). In all plots and statistical tests. Data are presented as mean ± s.e.m. ns = not significant

($p > 0.05$), *$p < 0.05$, **$p < 0.01$, ***$p < 0.001$, ****$p < 0.0001$. Sample sizes ($n$) are indicated in the figure panels. The synaptic interactions were visualized utilizing the GRASP system in male flies. **(I)** The synaptic interactions visualized utilizing the GRASP system in naïve and experienced male flies. The GFP fluorescence was processed using ImageJ software, where a threshold function was applied to distinguish fluorescence from the background. **(J)** Quantification of synaptic puncta formed between *SIFa^2A-lexA^* and *GAL4^24F06^* in brain between naïve and single male flies (two-tailed unpaired *t* test). In all plots and statistical tests. Data are presented as mean ± s.e.m. ns = not significant ($p > 0.05$), *$p < 0.05$, **$p < 0.01$, ***$p < 0.001$, ****$p < 0.0001$. Sample sizes ($n$) are indicated in the figure panels. The synaptic interactions were visualized utilizing the GRASP system in male flies. **(K)** Quantification of average synapse size formed between *SIFa^2A-lexA^* and *GAL4^24F06^* in brain between naïve and experienced male flies (two-tailed unpaired *t* test). In all plots and statistical tests. Data are presented as mean ± s.e.m. ns = not significant ($p > 0.05$), *$p < 0.05$, **$p < 0.01$, ***$p < 0.001$, ****$p < 0.0001$. Sample sizes ($n$) are indicated in the figure panels. The synaptic interactions were visualized utilizing the GRASP system in male flies. **(L)** No synapses were formed between *SIFa^2A-lexA^* and *GAL4^24F06^* in VNC. Scale bars represent 50 μm. Underlying data for all graphs can be found in file S1 Data.
(TIF)

**S5 Fig. Social context modulates both the formation of synapses between *GAL4^SIFa.PT^* and *lexA^24F06^* neurons in VNC and the calcium-dependent activity of *SIFaR*. (A–E)** Quantification of synaptic relative intensity formed between *GAL4^SIFa.PT^* and *lexA^SIFaR.2A^* in Brain and VNC between (B, D) naïve and single male flies; (C, E) naïve and experienced male flies (two-tailed unpaired *t* test). In all plots and statistical tests. Data are presented as mean ± s.e.m. ns = not significant ($p > 0.05$), *$p < 0.05$, **$p < 0.01$, ***$p < 0.001$, ****$p < 0.0001$. Sample sizes ($n$) are indicated in the figure panels. The synaptic interactions were visualized utilizing the tGRASP system in naïve, single and experienced male flies. Synaptic transmission occurs from *GAL4^SIFa.PT^* to *lexA^SIFaR.2A^*. **(F)** Different levels of neural activity of the brain as revealed by the CaLexA system in naïve, single and experienced flies. Male flies expressing *GAL4^24F06^* along with *LexAop-CD2-GFP, UAS-mLexA-VP16-NFAT and LexAop-CD8-GFP-A2-CD8-GFP* were dissected after 5 days of growth (mated male flies had 1-day of sexual experience with virgin females). The dissected brains were then immunostained with anti-GFP (green) and anti-nc82 (blue). GFP is pseudo-colored as "red hot". Boxes indicate the magnified regions of interest presented in the bottom panels. Scale bars represent 100 μm in brain and VNC panels. **(G–L)** Quantification of relative intensity value for GFP fluorescence (two-tailed unpaired *t* test). In all plots and statistical tests. Data are presented as mean ± s.e.m. ns = not significant ($p > 0.05$), *$p < 0.05$, **$p < 0.01$, ***$p < 0.001$, ****$p < 0.0001$. Sample sizes ($n$) are indicated in the figure panels. Underlying data for all graphs can be found in file S1 Data.
(TIF)

**S6 Fig. Social context modulates the formation of synapses between *GAL4^SIFa.PT^* and *lexA^SIFaR.2A^* neurons in CNS.**
**(A)** tGRASP assay for *GAL4^SIFa.PT^* and *lexA^SIFaR-2A^* in PI, PRW, OL and VNC region of male flies. Male flies expressing *GAL-4^SIFa.PT^* and *lexA^SIFaR-2A^* and *LexAop-2-pre-t-GRASP, UAS-post-t-GRASP* were dissected after 5 days of growth. Brains of male flies were immunostained with anti-GFP (green) and anti-nc82 (blue) antibodies. GFP is pseudo-colored as "red hot". Boxes indicate the magnified regions of interest presented in the bottom panels. Scale bars represent 100 μm in VNC panels, 25 μm in PI and PRW panels, and 50 μm in OL panels. Synaptic transmission occurs from *lexA^SIFaR-2A^* to *GAL4^SIFa.PT^*. **(B–I)** Quantification of synaptic relative intensity formed between *GAL4^SIFa.PT^* and *lexA^SIFaR-2A^* in (J) brain, (L) PI, (N) PRW and (P) OL between naïve and single male flies (two-tailed unpaired *t* test). In all plots and statistical tests. Data are presented as mean ± s.e.m. ns = not significant ($p > 0.05$), *$p < 0.05$, **$p < 0.01$, ***$p < 0.001$, ****$p < 0.0001$. Sample sizes ($n$) are indicated in the figure panels. The same quantification was performed for the relative synaptic intensity in these regions between naïve and experienced male flies. The synaptic interactions were visualized utilizing the tGRASP system in naïve, single and experienced male flies. **(J)** Diagram of differential SIFa-SIFaR signaling across various regions of the CNS in male *Drosophila melanogaster*, contingent upon diverse social contexts. **(K)** Schematic representation of neuronal

circuits modulating CNS internal states through feed-forward augmentation in response to social contexts. **(L)** Feed-forward enhancement circuits of *SIFa-SIFaR-SIFa*. Underlying data for all graphs can be found in file S1 Data.
(TIF)

**S7 Fig. *Crz* signaling *via SIFaR* predominantly shapes SMD behavior. (A–D)** Quantification of relative intensity value for GFP fluorescence in brain (A, B), and VNC (C, D) (two-tailed unpaired $t$ test). In all plots and statistical tests. Data are presented as mean ± s.e.m. ns = not significant ($p > 0.05$), $*p < 0.05$, $**p < 0.01$, $***p < 0.001$, $****p < 0.0001$. Sample sizes ($n$) are indicated in the figure panels. **(E)** Male flies VNC expressing *Crz-GAL4* and *lexA24F06* drivers together with *UAS-mCD8RFP* and *lexAop-mCD8GFP* were immunostained with anti-GFP (green), anti-DsRed (red), and anti-nc82 (blue) antibodies. Scale bars represent 100 μm in VNC panels and 50 μm in AG panels. Boxes indicate the magnified regions of interest presented in the bottom panels. The panels presented as a gray scale are to clearly show the nucleus in the adult VNC and AG labeled by *Crz-GAL4* and *lexA24F06* driver. **(F)** Male flies VNC expressing *Crz-GAL4* and *lexA24F06* drivers together with *UAS-Stinger* and *lexAop-tdTomato.nls* were immunostained with anti-GFP (green), anti-DsRed (red), and anti-nc82 (blue) antibodies. Scale bars represent 50 μm. Boxes indicate the magnified regions of interest presented in the bottom panels. The upper panels are presented as a gray scale to clearly show the nucleus in the adult VNC and AG labeled by *Crz-GAL4* and *lexA24F06* driver. Green arrows indicate *Crz+* nucleus. **(G–I)** Crzs slightly respond to SIFas activity in OL. Fluorescence changes ($\Delta F/F_0$) of GCaMP7s in Crzs after optogenetic stimulation of SIFas (two-tailed unpaired $t$ test). In all plots and statistical tests. Data are presented as mean ± s.e.m. ns = not significant ($p > 0.05$), $*p < 0.05$, $**p < 0.01$, $***p < 0.001$, $****p < 0.0001$. Sample sizes ($n$) are indicated in the figure panels. LED light were fired 2 s after 30 s of dark. $N = 4$ in each group. Scale bars represent 50 μm. Underlying data for all graphs can be found in file S1 Data.
(TIF)

**S8 Fig. Non-neuronal cells required for SIFa-SIFaR/Crz-CrzR signaling in interval timing behaviors. (A)** Dot plot depicting the expression levels of *CrzR* in 'adult oenocyte' as annotated by the Fly Cell Atlas (FCA) within the oenocyte tissue. The gene expression levels for individual cells were normalized using the 'LogNormalize' method with a scale factor of 10,000, followed by scaling of all genes. See the Materials and methods for a detailed description of the single-nucleus RNA-sequencing analyses used in this study. **(B)** Each tSNE visualization depicts the coexpression patterns of genes, with each color corresponding to the genes listed on the left, right, and bottom of the plot. The tissue name, as referenced on the Fly SCope website is indicated in the upper left corner of the tSNE plot. See the Materials and methods for a detailed description of the single-nucleus RNA-sequencing analyses used in this study. **(C, D)** LMD and SMD assays for *PromE-GAL4* mediated knockdown of SIFaR *via SIFaR-RNAi* together with *tub-GAL80ts* (two-tailed unpaired $t$ test). In all plots and statistical tests. Data are presented as mean ± s.e.m. ns = not significant ($p > 0.05$), $*p < 0.05$, $**p < 0.01$, $***p < 0.001$, $****p < 0.0001$. Sample sizes ($n$) are indicated in the figure panels. **(E, F)** LMD and SMD assays for *elavc155* (neuron) -mediated knockdown of CrzR *via CrzR-RNAi* (two-tailed unpaired $t$ test). In all plots and statistical tests. Data are presented as mean ± s.e.m. ns = not significant ($p > 0.05$), $*p < 0.05$, $**p < 0.01$, $***p < 0.001$, $****p < 0.0001$. Sample sizes ($n$) are indicated in the figure panels. **(G, H)** LMD and SMD assays for *repo-GAL4* (glia) -mediated knockdown of CrzR *via CrzR-RNAi* (two-tailed unpaired $t$ test). In all plots and statistical tests. Data are presented as mean ± s.e.m. ns = not significant ($p > 0.05$), $*p < 0.05$, $**p < 0.01$, $***p < 0.001$, $****p < 0.0001$. Sample sizes ($n$) are indicated in the figure panels. **(I)** The results of qRT-PCR for SIFaR gene expression. The gray bar represents the control group, which has the genotype *+/CrzR-RNAi*. The red bar indicates the experimental group with the genotype *tub-GAL4; tub-GAL80ts/CrzR-RNAi*. The y-axis depicts the relative expression level of SIFaR, normalized to the expression of the GAPDH gene. "RGX" denotes relative gene expression. **(J-O)** Genetic control assays were performed using heterozygous *PromE-GAL4, tub-GAL80ts/+*, *CrzR-RNAi/+*, and *repo-GAL4/+* males (two-tailed unpaired $t$ test). In all plots and statistical tests. Data are presented as mean ± s.e.m. ns = not significant ($p > 0.05$), $*p < 0.05$, $**p < 0.01$,

***$p < 0.001$, ****$p < 0.0001$. Sample sizes ($n$) are indicated in the figure panels. Underlying data for all graphs can be found in file S1 Data.
(TIF)

**S1 Table. Summary of SlFaR(24F06)-GAL4 mediated neuropeptide knockdown results.** N/A cells show that males are unable to effectively mate with females for 1 h during the mating duration assay. *t-value* indicate the t-statistic. *df-value* indicate the degrees of freedom. *p-value* marked in color indicate the disruption of LMD (blue) or SMD (red). Sample sizes ($n$) are indicated in the table.
(DOCX)

**S2 Table. Summary of NP-GAL4 mediated *SlFaR* knockdown results.** *t-value* indicate the t-statistic. *df-value* indicate the degrees of freedom. *p-value* marked in color indicate the disruption of LMD (blue) or SMD (red). Sample sizes ($n$) are indicated in the table.
(DOCX)

**S1 Supplemental information. Identity of GAL4$^{24F06}$.** **(A)** Male flies brain expressing *GAL4$^{24F06}$* and *fru$^{FLP}$* together with *UAS-stop-mCD8GFP* were immunostained with anti-GFP (green) and anti-nc82 (blue) antibodies. Scale bars represent 100 µm in brain and 50 µm in VNC panels. **(B)** Male flies brain expressing *GAL4$^{24F06}$* and *dsx$^{FLP}$* together with *UAS-stop-mCD8GFP* were immunostained with anti-GFP (green) and anti-nc82 (blue) antibodies. Scale bars represent 100 µm in brain and 50 µm in VNC panels. **(C)** Male flies brain expressing *GAL4$^{AstA-2A}$* and *lexA$^{24F06}$* drivers together with *UAS-mCD8RFP* and *lexAop-mCD8GFP* were immunostained with anti-GFP (green), anti-DsRed (red), and anti-nc82 (blue) antibodies. Dashed circles indicate the regions of interest. Scale bars represent 50 µm in SOG and 20 µm in AG panel. **(D)** Male flies brain expressing *GAL4$^{Capa-2A}$* and *lexA$^{24F06}$* drivers together with *UAS-mCD8RFP* and *lexAop-mCD8GFP* were immunostained with anti-GFP (green), anti-DsRed (red), and anti-nc82 (blue) antibodies. Dashed circles indicate the regions of interest. Scale bars represent 50 µm in SOG and 20 µm in AG panel. **(E)** Male flies brain expressing *GAL4$^{FMRFa-2A}$* and *lexA$^{24F06}$* drivers together with *UAS-mCD8RFP* and *lexAop-mCD8GFP* were immunostained with anti-GFP (green), anti-DsRed (red), and anti-nc82 (blue) antibodies. Dashed circles indicate the regions of interest. Scale bars represent 50 µm in SOG and 20 µm in AG panel. **(F)** Male flies brain expressing *GAL4$^{Lk-2A}$* and *lexA$^{24F06}$* drivers together with *UAS-mCD8RFP* and *lexAop-mCD8GFP* were immunostained with anti-GFP (green), anti-DsRed (red) and anti-nc82 (blue) antibodies. Dashed circles indicate the regions of interest. Scale bars represent 50 µm in SOG and 20 µm in AG panel. **(G)** Male flies brain expressing *GAL4$^{MS-2A}$* and *lexA$^{24F06}$* drivers together with *UAS-mCD8RFP* and *lexAop-mCD8GFP* were immunostained with anti-GFP (green), anti-DsRed (red), and anti-nc82 (blue) antibodies. Dashed circles indicate the regions of interest. Scale bars represent 50 µm in SOG and 20 µm in AG panel. **(H)** Male flies brain expressing *GAL4$^{Proc}$* and *lexA$^{24F06}$* drivers together with *UAS-mCD8RFP* and *lexAop-mCD8GFP* were immunostained with anti-GFP (green), anti-DsRed (red), and anti-nc82 (blue) antibodies. Dashed circles indicate the regions of interest. Scale bars represent 50 µm.
(TIF)

**S2 Supplemental information. Raw data of SlFa-PT-to-SlFaR-2A-tGRASP.**
(ZIP)

**S3 Supplemental information. Raw data of GAL4$^{24F06}$-CalexA.**
(ZIP)

**S1 Data. Underlying numerical data for graphs in** Figs 1A–1K; 2B–2K; 3C–3J; 4C–4G, 4J–4O; 5B–5G, 5J, 5K; 6C–6O; 7B–7G; S1A–S1R; S2A–S2F; S3C, S3E; S4D, S4E, S4G, S4H, S4J, S4K; S5B–S5E, S5G–S5L; S6B–S6I; S7A–S7D, S7G, S7H; S8C–S8O.
(XLSX)

## Acknowledgments

We thank Dr. Jan A. Veenstra (University of Bordeaux) for sharing *SIFa*$^{PT}$*-GAL4* driver, Dr. Young-Joon Kim (GIST) for kindly sharing *UAS-SIFaR* fly strain, Drs. Yuh Nung Jan and Lily Yeh Jan (UCSF, USA) for helpful comments and support on this paper. We are also very appreciative to the colleagues who supplied us with several fly strains: Dr. Wei Zhang (Tsinghua University), Fang Guo (Zhejiang University), and Dr. Yufeng Pan (Southeast University) and Drs. Young-Joon Kim and Sung-Eun Yoon (Korea *Drosophila* Resource Center, KDRC). We have greatly benefited from the resources provided by the FlyBase website in our genetic research endeavors. We are grateful for the ongoing efforts of the FlyBase staff in maintaining this comprehensive *Drosophila* database.

## Statement of animal research compliance

All animal experiments reported in this manuscript were conducted in compliance with the ARRIVE guidelines and adhered to the U.K. Animals (Scientific Procedures) Act, 1986 and associated guidelines, EU Directive 2010/63/EU for animal experiments, or the National Research Council's Guide for the Care and Use of Laboratory Animals.

## Declaration of generative AI and AI-assisted technologies in the writing process

During the creation of this work, the author(s) utilized QuillBot to rephrase English sentences, verify English grammar, and detect plagiarism, as none of the authors of this paper are native English speakers. After using this tool/service, the author(s) reviewed and edited the content as needed and take(s) full responsibility for the content of the publication.

## Author contributions

**Conceptualization:** Woo Jae Kim.

**Data curation:** Tianmu Zhang, Zekun Wu, Yutong Song, Tae Hoon Ryu, Kyle Wong, Justine Schweizer, Woo Jae Kim.

**Formal analysis:** Tianmu Zhang, Zekun Wu, Yutong Song, Tae Hoon Ryu, Xiaoli Zhang, Wenjing Li, Yanying Sun, Kyle Wong, Justine Schweizer, Khoi-Nguyen Ha Nguyen, Alex Kwan, Woo Jae Kim.

**Funding acquisition:** Woo Jae Kim.

**Investigation:** Woo Jae Kim.

**Methodology:** Woo Jae Kim.

**Project administration:** Woo Jae Kim.

**Resources:** Woo Jae Kim.

**Software:** Woo Jae Kim.

**Supervision:** Kweon Yu, Woo Jae Kim.

**Validation:** Tianmu Zhang, Zekun Wu, Woo Jae Kim.

**Visualization:** Zekun Wu, Woo Jae Kim.

**Writing – original draft:** Woo Jae Kim.

**Writing – review & editing:** Tianmu Zhang, Woo Jae Kim.

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
