## [Editor Report · Decision Letter 0]

18 Dec 2024

Dear Dr Kim, 

As mentioned in my last email, which was sent on your co-submitted paper "Peptidergic neurons with extensive branching orchestrate the internal states and energy balance of male Drosophila melanogaster", I am writing to let you know that we would like to send your PLOS Biology manuscript "Long-range neuropeptide relay as a mechanism for context-dependent interval timing behaviors", out for review. We are inviting you to complete the metadata for this study as well, as this is needed before we can contact the reviewers. 

As detailed in my previous email, I have now had a chance to discuss this study, which was reviewed at Review Commons and then re-reviewed at EMBO J, with one of our Academic Editors. In light of the changes made in this most recent revision, we would like to send the study back to the original reviewers to see if they are satisfied with the changes made. 

Before we can send your manuscript to reviewers, we need you to complete your submission by providing the metadata that is required for full assessment. To this end, please login to Editorial Manager where you will find the paper in the 'Submissions Needing Revisions' folder on your homepage. Please click 'Revise Submission' from the Action Links and complete all additional questions in the submission questionnaire.

Once your full submission is complete, your paper will undergo a series of checks in preparation for peer review. After your manuscript has passed the checks it will be sent out for review. To provide the metadata for your submission, please Login to Editorial Manager (https://www.editorialmanager.com/pbiology) within two working days, i.e. by Dec 20 2024 11:59PM.

Kind regards,

Luke

Lucas Smith, Ph.D.

Senior Editor

PLOS Biology

lsmith@plos.org

---

## [Decision Letter · Decision Letter 1]

7 Mar 2025

Dear Woo Jae, 

Thank you for your patience while your manuscript "Long-range neuropeptide relay as a mechanism for context-dependent interval timing behaviors" was peer-reviewed at PLOS Biology. We understand that this study has been revised in response to reviews from Review Commons and then also in response t a second round of reviews from EMBO J. I apologize again for the protracted review process for this study as well. As with your co-submitted paper, PBIOLOGY-D-24-03577R1, the original reviewers 1 and 2 were not available to assess this study, and so we ended up recruiting a new, reviewer 4 to help assess the revision. Reviewer 4 is the same for both of your papers. In this case, we did not reach out to the original reviewer 3 for this study, as s/he previously was satisfied by the revision when it was submitted to EMBO J. 

Reviewer 4’s comments are appended below. While s/he appreciates the amount of data provided here, you will see that reviewer 4 has a number of lingering concerns with the presentation and conclusiveness of this study, which relate to comments from the previous round of review. 

We have discussed these concerns with the Academic Editor – and as with your related paper, we think that the study is generally easy enough to follow and so would not require that you majorly restructure the piece. However, given reviewer 4’s feedback, we do think the manuscript would benefit from an edit focused on clarity, and that the specific examples highlighted by reviewer 4 should be addressed. We also encourage you to include the relevant genetic controls in the main figures – although again, if those were generated independently that should be clearly indicated. 

In addition to their comments about the presentation of the study – reviewer 4 does also raise concerns with the conclusiveness of some of the data. While we appreciate that some of these points can be addressed with textual changes and adjusting conclusions, we think that additional data should be provided where requested. We think in places, more work will be needed to increase the sample size, and that you should carefully respond to concerns raised by reviewer 4 about the functional connectivity GCAMP data – and this point may require additional analyses to address, too. 

I also note that reviewer 4 has questioned whether FlyLight images can be used without prior approval – and we ask that you confirm, in your cover letter, that these are indeed available for you to publish, under an open-access CC-By license. For images that were generated elsewhere, the figure legend should clearly indicate this, ideally with a reference. For example the figure legend for Flylight data should read “Reproduced from Fig xx of REFERENCE”. 

Given our interest in this study, we are willing to give you one last chance to address these issues in another round of revision. However, given the manuscript’s review history, I should also note that this is the last round of experimental revision that we will entertain and that we will be looking to see that the reviewer concerns are adequately addressed, including with new data, before we can consider your paper for publication at PLOS Biology. 

As with your other paper, if it would be helpful, we are happy to discuss and provide input on a revision plan for this study. 

**IMPORTANT - SUBMITTING YOUR REVISION**

*Re-submission Checklist*

*Published Peer Review*

*PLOS Data Policy*

*Blot and Gel Data Policy*

Sincerely,

Luke

Lucas Smith, Ph.D.

Senior Editor

PLOS Biology

lsmith@plos.org

REVIEWS:

New Reviewer #4 (who assessed the responses to the original reviewers from Review Commons): Title: Long-range neuropeptide relay as a mechanism for context-dependent interval timing behaviors

Major comments: 

This manuscript by Zhang et al explores the role of SIFa signaling in regulating mating duration in Drosophila. I have looked at the comments of the reviewers on the previous version of this manuscript and assessed the revised manuscript. While I commend the authors on the different suites of methods used in this manuscript, I generally shared the concerns of the previous reviewers. Despite the revisions, it is still difficult to follow the logic behind a lot of the experiments and the conclusions from them are sometimes not supported. Below are the specific major issues with this manuscript:

* The writing is difficult to understand in a lot of places and I had to extrapolate what the authors were trying to say. I feel that the writing is not at the level expected for PLoS Biology. I provide some specific examples below. 

o "Interorgan communication is also represented by neuromodulatory signaling, which is even hormonal and conducted via the circulation." 

o "The modulating energy homeostasis between the gut and the brain is one of the most intensively studied neuromodulatory circuits via the neuronal relay of neuropeptides" 

o "The relayed event of non-synaptic and diffuse neurotransmission through neuropeptides and their receptors is also essential for the maintenance of metabolic homeostasis in Drosophila"

o "However, the mechanism by which SIFa signaling controls various behaviors through its receptor SIFaR is not well known, because it is difficult to resolve using ordinary molecular genetics approaches and typical neural connectomics may not necessarily predict peptidergic circuits." - the authors state that it is difficult to do this and then go and use genetics tools for synaptic tracing and connectomics to find synaptic partners. 

* The presentation of the figures is still an issue. It is difficult to find the relevant genetic controls because they are all combined in one figure. The presentation of the confocal images (use of colors, thresholding, multiple reporters) is also questionable. 

* While the authors narrowed down the neurons (24F06-GAL4) that could be mediating the effects of SIFa, the precise identity of neurons expressing SIFaR important for this behavior is still lacking. 

* The functional connectivity data using GCAMP are not convincing due to the decay in signal in some controls but not others. Calcium signals should not drop this drastically within the 60 seconds presented in the figures. 

Specific comments

Line 100: SIFa is expressed in more than 4 neurons and includes neurons that innervate the central complex. It does have strongest expression in the 4 neurons. 

Figure S1A: How can prior exposure to competing males be regarded as naive males? Shouldn't it be referred to as grouped to distinguish from single?

Line 130-133: "The SIFaR, the receptor for SIFa, has been linked to circadian rhythm, feeding, courtship, sleep, and memory extinction; however, its functional properties remain poorly understood" . If those are the functions of SIFaR then how are its "functional properties" poorly understood? 

Line 141-142: It is extremely difficult to find the relevant genetic controls for the experiments presented in Figure 1A-D. Supplemental information. 1 is just one figure containing genetic controls from different experiments. I dont understand why the authors have chosen this format for presenting the data. 

Figure 1F-G: It is extremely difficult to make any sense of the SIFaR expression pattern based on current presentation. Compared to the staining pattern for the same driver depicted in the chemoconnectome manuscript, none of the images add anything new. 

Lines 173-178: Previous reviewers have all critiqued the interpretation of nc82 and GFP staining. I agree with them. The authors over interpreting their findings. The only conclusion that can be made from their staining is that SIFaR is broadly expressed in the CNS. 

Line 184: replace 2promoter" with "regulatory" since two drivers target an intronic region which is not normally a "promoter". 

Line 187-189: "In the spatial, the targeted reduction of SIFaR expression in the GAL424F06 neuronal subset resulted in a notable alteration of mating behavior." - I dont understand what in the spatial means. 

Figure 2 results: The whole discussion on disinhibition is long, speculative and out of place in the results section. 

Figure 2F: What is the point of having greyscale and colored images for the same staining? The reporter in the figure is red stinger but the images are of GFP staining from FlyLight. I am not sure if FlyLight images can be used in publication without approval. I also dont understand the thresholding in Figure 2G. Why is not sufficient to just show the staining pattern in males and females without commenting on differences in numbers which were not properly quantified? 

Figure S2G: It would have been much informative if the authors had used a FLP based fluorescence reporter to show only the common neurons between the two lines. At the moment it is not possible to see any detailed morphology about the neurons that are common in both drivers. 

Line 231.243: The switch from LMD/SMD to feeding does not make sense and it is out of place. It is just results that dont fit in with the rest of the story.

Line 247-260: As mentioned by previous reviewers, I also dont understand the reasoning for doing this kind of colocalization analysis. Not all 24F06-GAL4 neurons express SIFaR so I am not sure what this analysis adds. For the same reasons, I think that conclusions based on Figure 3 are premature - the authors have not shown that all the 24F06-GAL4 neurons are SIFaR positive. Therefore, synapses could be formed by neurons that dont express SIFaR. The images in Figure S3D are too small. 

Figure S3F and G: The authors claim that they used the FlyWire connectome to do the analyses but the IDs provided are for SIFa neurons in hemibrain which is a completely different dataset. Please clarify what dataset was actually used for the analyses. 

Figure 4A-H. What is the expression pattern of the GAL4 used in these experiments? My issue with GRASP is that if SIFaR expression is altered under different contexts, it will directly affect the amount of reporter signal since this is a LexA knock-in line. This could affect the GRASP signal even if the number of synapses remains constant. 

Figure 4P: The schematic needs to be altered as not all 24F06-GAL4 neurons express SIFaR. Hence, some of these cells could also be activated by other neuropeptides/neurotransmitters besides SIFa. 

Figure 5H-L: There are several neurons in the AG that are labelled by the 24F06 GAL4. In fact, most of the AG seems stained with GFP in Figure 2F. So which neurons was GCAMP signal measured in? The representative figure in Figure 5L seems to show very few pixels and the number of neurons cannot be estimated from that image. I also share the concern with the previous reviewer that the calcium signal in Figure 5J should not decay so fast (within a minute) in the control. I am not sure if this is because the cell was not functioning normally. 

Lines 405-406: It is known that no Crz peptide is produced in the OL. This is due to post-transcriptional regulation of Crz mRNA. (https://pubmed.ncbi.nlm.nih.gov/18087727/). 

Figure 6K-L: sample size of 3 is not sufficient. The decay in calcium signal in control is an issue again especially since the control in Figure 6N looks completely stable over the imaging period. 

Lines 443-445: This conclusion is not fully supported by the data for the reasons highlighted above.

---

## [Editor Report · Decision Letter 2]

3 Jul 2025

Dear Woo Jae,

Thank you for your patience while we considered your revised manuscript "Long-range neuropeptide relay as a mechanism for context-dependent interval timing behaviors" for publication as a Research Article at PLOS Biology. This revised version of your manuscript has been evaluated by the PLOS Biology editors and the Academic Editor who is fully satisfied by your response to reviewers, and who suggests that we accept your study. 

We agree that you have done a very good job addressing the reviewer comments, and based on our Academic Editor's assessment of your revision we are likely to accept this manuscript for publication. However please not that before we can editorially accept your study, we need you to address a few remaining data and other policy-related requests, in a last, short revision. These are detailed below. 

**IMPORTANT: please address the following editorial requests**

1) TITLE: We would suggest that you edit the title to reflect more clearly the specific findings from your study. If you agree, we suggest you change it to something like: 

"Neuropeptide-mediated synaptic plasticity regulates context-dependent mating behaviors in Drosophila"

2) DATA: You may be aware of the PLOS Data Policy, which requires that all data be made available without restriction: http://journals.plos.org/plosbiology/s/data-availability. For more information, please also see this editorial: http://dx.doi.org/10.1371/journal.pbio.1001797

a. Supplementary files (e.g., excel). Please ensure that all data files are uploaded as 'Supporting Information' and are invariably referred to (in the manuscript, figure legends, and the Description field when uploading your files) using the following format verbatim: S1 Data, S2 Data, etc. Multiple panels of a single or even several figures can be included as multiple sheets in one excel file that is saved using exactly the following convention: S1_Data.xlsx (using an underscore).

b. Deposition in a publicly available repository. Please also provide the accession code or a reviewer link so that we may view your data before publication. 

>>Regardless of the method selected, please ensure that you provide the individual numerical values that underlie the summary data displayed in the following figure panels as they are essential for readers to assess your analysis and to reproduce it:

Fig 1A-K; Fig 2A-K; Fig 3C-J; Fig 4B-H,J-O; Fig 5B-G,J-K; Fig 6 C-O; Fig 7B-G;

Fig S1A-K; Fig S2A-F; Fig S3 E; Fig S4 D-E,G-h,J-K; Fig S5 B-E,G-L; Fig S6B-I; Fig S7A-D,G-H; Fig S8 A-O;

>>Please also ensure that figure legends in your manuscript include information on where the underlying data can be found, and ensure your supplemental data file/s has a legend.

>>Please ensure that your Data Statement in the submission system accurately describes where your data can be found.

3) CODE: Per journal policy, if you have generated any custom code during the course of this investigation, please make it available without restrictions. Please ensure that the code is sufficiently well documented and reusable, and that your Data Statement in the Editorial Manager submission system accurately describes where your code can be found.

4) RELATED MANUSCRIPT: Based on our recent email correspondence, I understand that you plan to submit the revised version of your related manuscript soon. Would you like us to try to co-publish your two papers (assuming that we find the other paper suitable for publication as well)? That would likely require us to delay the publication of this paper... Or do you prefer that we uncouple the two studies, and publish this as soon as possible? (once the last editorial requests are addressed). Please add a note in the cover letter, indicating your preference. 

We expect to receive your revised manuscript within two weeks. 

*Published Peer Review History*

*Press*

Sincerely,

Luke

Lucas Smith, Ph.D.

Senior Editor

lsmith@plos.org

PLOS Biology

---

## [Editor Report · Decision Letter 3]

22 Jul 2025

Dear Woo Jae,

Thank you for the submission of your revised Research Article "Neuropeptide-mediated synaptic plasticity regulates context-dependent mating behaviors in Drosophila" for publication in PLOS Biology and thank you for addressing our last editorial requests in this revision. On behalf of my colleagues and the Academic Editor, Paul J Shaw, I am pleased to say that we can in principle accept your manuscript for publication, provided you address any remaining formatting and reporting issues. These will be detailed in an email you should receive within 2-3 business days from our colleagues in the journal operations team; no action is required from you until then. Please note that we will not be able to formally accept your manuscript and schedule it for publication until you have completed any requested changes.

PRESS

Sincerely, 

Luke

Lucas Smith, Ph.D.

Senior Editor

PLOS Biology

lsmith@plos.org